# Single-cell chromatin accessibility and transcriptomic characterization of Behcet's disease

Wen Shi[1,2], Jinguo Ye[1], Zhuoxing Shi[1], Caineng Pan[1], Qikai Zhang[1], Yuheng Lin[1], Dan Liang [1✉], Yizhi Liu [1,2✉], Xianchai Lin [1,2✉] & Yingfeng Zheng [1,2✉]

Behect's disease is a chronic vasculitis characterized by complex multi-organ immune aberrations. However, a comprehensive understanding of the gene-regulatory profile of peripheral autoimmunity and the diverse immune responses across distinct cell types in Behcet's disease (BD) is still lacking. Here, we present a multi-omic single-cell study of 424,817 cells in BD patients and non-BD individuals. This study maps chromatin accessibility and gene expression in the same biological samples, unraveling vast cellular heterogeneity. We identify widespread cell-type-specific, disease-associated active and pro-inflammatory immunity in both transcript and epigenomic aspects. Notably, integrative multi-omic analysis reveals putative TF regulators that might contribute to chromatin accessibility and gene expression in BD. Moreover, we predicted gene-regulatory networks within nominated TF activators, including AP-1, NF-kB, and ETS transcript factor families, which may regulate cellular interaction and govern inflammation. Our study illustrates the epigenetic and transcriptional landscape in BD peripheral blood and expands understanding of potential epigenomic immunopathology in this disease.

[1] State Key Laboratory of Ophthalmology, Zhongshan Ophthalmic Center, Sun Yat-sen University, Guangdong Provincial Key Laboratory of Ophthalmology and Visual Science, 510060 Guangzhou, China. [2] Research Unit of Ocular Development and Regeneration, Chinese Academy of Medical Sciences, 100085 Beijing, China. ✉email: liangdan@gzzoc.com; liuyizh@mail.sysu.edu.cn; linxch7@mail.sysu.edu.cn; zhyfeng@mail.sysu.edu.cn

Behcet's disease (BD) is a systemic inflammatory disorder of unknown etiology affecting blood vessels[1]. It commonly manifests as inflammation of the intra-ocular structure, and recurrent oral/genital ulceration. BD causes morbidity and mortality, particularly in Asians[2]. Current treatments for BD are aggressive systemic and topical glucocorticoids, with or without immunosuppressive agents. However, these can lead to undesirable side effects, such as hyperglycemia, osteoporosis, and obesity, related to prolonged drug usage[3]. Thus, there is a need to develop new targeted therapies for BD. Painful skin lesions, recurrent ulceration, and blindness result from the combination of genetic susceptibility, environmental triggers, and dysregulated immune responses involving T helper 17 (Th17) cells, monocytes, skin CD8+ T cells and pro-inflammatory cytokines[4–8]. However, to date, the knowledge of genetic contributors and pathogenic cells to BD is still limited.

Over recent decades, progress in single-cell sequencing technologies has enabled profiling of the genetic transcriptomics of peripheral blood mononuclear cells (PBMCs) and skin tissues from BD patients[4,8]. Although previous studies have examined the single-cell gene expression of BD, heterogeneity in the single-cell epigenomics of PBMCs has not been profiled. Integrating the single-cell assay for transposase-accessible chromatin sequencing (scATAC-seq) and single-cell RNA-seq (scRNA-seq) enables the identification of the potential disease-associated regulatory program[9–16].

In this study, we aimed to map the cellular landscape of PBMCs in BD patients, with the goal of dissecting disease heterogeneity among patients and identify the underlying cellular and molecular events. To accomplish this, we simultaneously generated both transcriptomic and epigenomic data in BD patients to identify the gene regulatory network. Our analyses uncovered widespread gene expression and chromatin accessibility changes in both BD patients and unaffected controls, including hyperactivation signatures in T cells and monocytes. Notably, we also nominated potential TF activators of chromatin accessibility and gene expression in BD. Moreover, our multi-omics analysis was effective at predicting disease regulatory networks, highlighting the predicted involvement of AP-1, NF-kB, and ETS transcript factor families in BD pathophysiology. Overall, our study provides insights into the understanding of the peripheral immune pathogenesis of BD.

## Results

### High-resolution single-cell epigenomic and transcriptional peripheral immune cell-type mapping of Behcet disease patients

We performed droplet-based scRNA-seq and scATAC-seq (10X Genomics) to map the immune landscape of PBMCs from 22 BD patients in scATAC-seq dataset, 23 BD patients in scRNA-seq dataset and 8 non-BD individuals in both dataset (Fig. 1a, Supplementary Data 1). After stringent quality control filtration, a total of 152,704 cells of the scATAC-seq dataset and 272,113 cells of the scRNA-seq dataset were retained for downstream analysis, with an average of 8810 unique nuclear fragments and an average of 14.5 in TSS enrichment for scATAC-seq-profiled cells, and an average of 2042.9 UMIs for scRNA-seq-profiled cells (Fig. 1b, Supplementary Fig. 1a–h). We did not detect any potential batch effects in our datasets (Supplementary Fig. 1d–g). Therefore, no batch correction method was applied in our further analysis. The quality control thresholds of the scATAC-seq and scRNA-seq are described in the Methods. The scATAC-seq dataset, aligned using dimension reduction and graph-based clustering, yielded discrete cell clusters, primarily representing T (CD4/CD8) cells, monocytes, dendritic cells (DCs), T cells, natural killer (NK) cells, B lymphocytes, and Hematopoietic stem and progenitor cell (HSPC) (Fig. 1c,

Supplementary Data 2). With scATAC-seq, we first manually annotated based on chromatin accessibility at the promoter regions of key lineage markers for six major immune cell lineages of the PBMCs by comparing differentially accessible chromatin regions (DARs): T cells (84,202 cells; 1–4, clusters 7–9); monocytes (29,624 cells; clusters 16–19); DCs (1,861 cells; clusters 11, 15); NKs (22,738 cells; clusters 5, 6); BCs (12,610 cells; clusters 10, 12, 13) and HSPC (1669 cells; 14 clusters)[17] (Fig. 1b–d). Open chromatin at known major immune cell lineages specific genes validated our analysis. T cells had high accessibility at cis-elements neighboring $CD8A$[17] and $IL7R$[8,17] (Fig. 1d, Supplementary Fig. 2a, c). NK cells had higher accessibility at $GNLY$[18]. Monocytes showed higher accessibility within $S100A8$[17] (Fig. 1d, Supplementary Fig. 2a, c). We found that HSPC had higher accessibility at $GATA2$[17] (Fig. 1d, Supplementary Fig. 2a, c). B cells had high accessibility at $MS4A1$[19] (Fig. 1d, Supplementary Fig. 2a, c). DCs showed higher accessibility within $HLA$-$DQA1$[20] (Fig. 1d, Supplementary Fig. 2a, c). We also used $chromVAR$[21] to compute transcription factor (TF) motif deviation in single cells by estimating the enrichment of TF binding motifs in open chromatin regions and examined the enrichment of TF motifs in immune cell types concerning diagnosis and identified NFKB1 TF motifs with increased enrichment with BD patients in monocytes (cluster 18, Fig. 1e, Supplementary Fig. 2b, d). For example, LEF1 was active in naive T cell lineage and myeloid cells shared the activity of SPI1 factor motif but demonstrated unique activity of the GATA2 factor in HSPC[17,22] (Fig. 1e, Supplementary Fig. 2b, d).

Likewise, we detected similar cell types and annotated them based on known marker genes using scRNA-seq: T cells (152,842 cells; clusters 1–3, 6, 7, 10, 11, 25); monocytes (53573 cells; clusters 5, 9, 12, 19, 21, 26); DCs (3000 cells; clusters 13, 14, 20); NKs (40875 cells; clusters 15, 17, 22); BCs (20971 cells; clusters 4, 8, 16, 18) and HSPC (852 cells; clusters 23, 24) (Fig. 1b, c, f). As expected, many differentially expressed genes (DEGs) in the six major immune cell types agreed with previous literature and our scATAC-seq dataset, such as $IL7R$ for T cells, $GNLY$ for NKs, $S100A8$ for monocytes, $HLA$-$DQA1$ for DC, $MS4A1$ for BC and $GATA2$ for HSPC[17,23] (Fig. 1f). Altogether, we mapped the high-resolution epigenomic and transcriptional peripheral immune landscape in BD at a single-cell resolution.

### Multi-omic characterization of T cell heterogeneity in BD blood

The high sensitivity of scATAC-seq and scRNA-seq allowed us to further map these major immune cell types into subtypes[22,23]. As in previous studies, we compared the DARs in the scATAC-seq dataset and DEGs in the scRNA-seq dataset to map T cell subsets[17,22]. We also used gene activity scores (GAS) for cell type identification in the scATAC-seq dataset due to the sparsity of single-cell cis-element information[24]. Specifically, we identified T cell subsets into 15 subgroups according to the expression of cell type marker GAS and genes (Fig. 2a, Supplementary Fig. 3a, Supplementary Fig. 4a, b), including CD4 cytotoxic T cells (CD4 CTLs; $CD4^+GZMK^+$), CD4 naïve T cells (CD4 Naïve; $CD4^+SELL^{++}TCF7^{++}$), CD4 central memory T cells (CD4 TCMs; $CD4^+SELL^+TCF7^+$), CD4 T follicular helper (CD4 TFH; $CD4^+ICOS^+CXCR5^+$), CD4 T helper 1 cells (CD4 Th1; $CD4^+CXCR3^+$), CD4 T helper 2 cells (CD4 Th2; $CD4^+CCR4^+$), CD4 T helper 17 cells (CD4 Th17; $CD4^+CCR6^+RORC^+$), CD4 regulatory T cells (CD4 Treg; $CD4^+FOXP3^+$), CD8 mucosal-associated invariant T cells (CD8 MAIT; $CD8^+CCR6^+RORC^+$), CD8 naïve T cells (CD8 Naïve; $CD8^+SELL^{++}TCF7^{++}$), CD8 central memory T cells (CD8 TCMs; $CD8^+SELL^+TCF7^+$), CD8 effector memory T cells (CD8 TEMs; $CD8^+IFNG^+GZMK^+$), CD8 exhausted T cells (CD8 TEXs; $CD8^+PDCD1^+$), CD8 regulatory T cells (CD8 Treg; $CD8^+KIR3DL2^+KIR2DL2^+$) and double negative T cells (DNT,

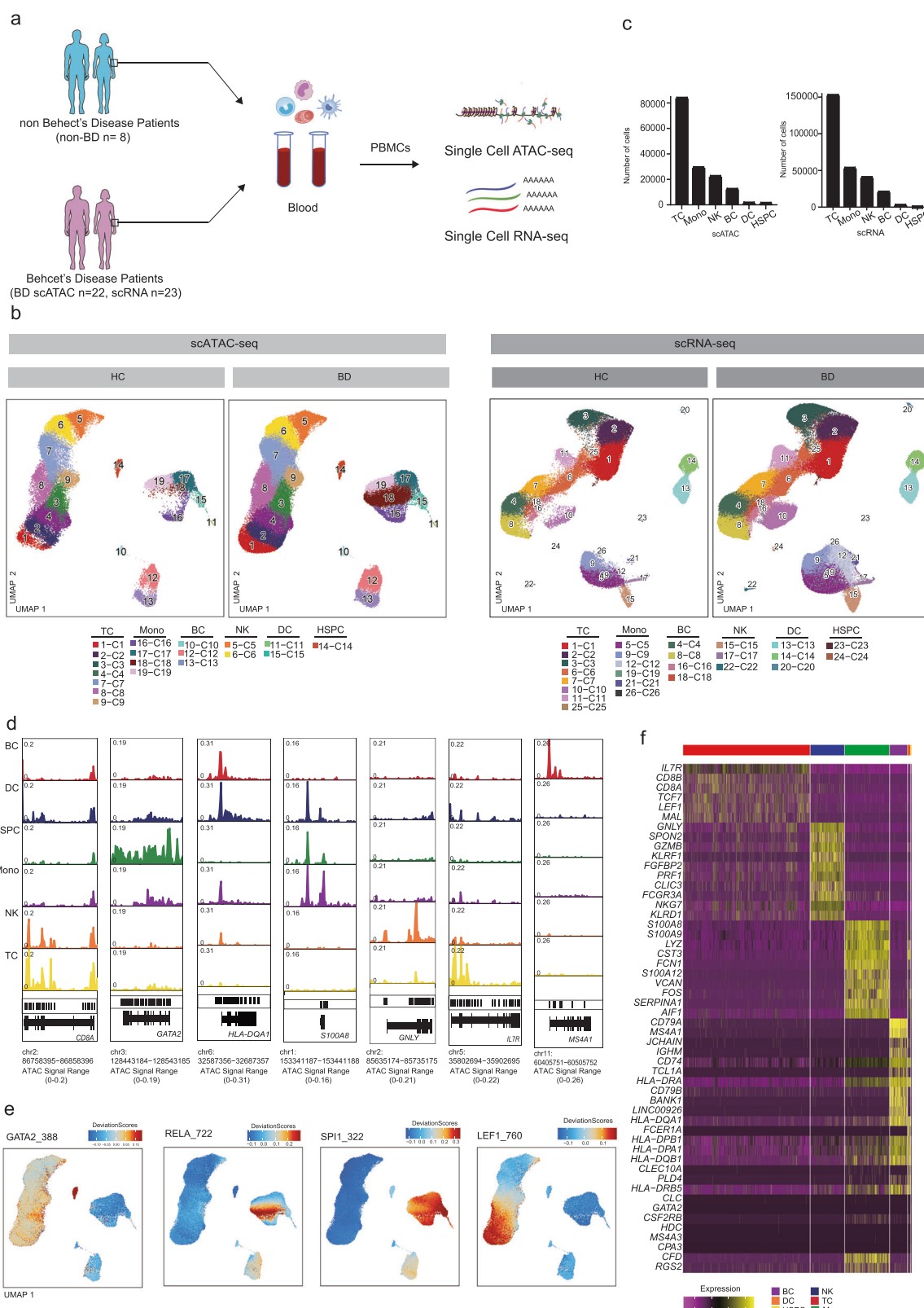

CD3+CD4+CD8−) (Fig. 2b, Supplementary Fig. 3a, Supplementary Fig. 4a, S4b)[17,22,23,25,26]. CD8 Tregs are rare cell types in PBMCs. Recently, it has been found that they undergo clonally expansion in some autoimmune disorders[27]. To further identify the regulatory TF in CD8 Treg, we compared the differentially accessible chromatin regions (DARs) and analyzed their enrichment of key TFs. Compared to CD8 naïve T cells, TF

Eomesodermin (EOMES) were the top TFs enriched in CD8 Tregs[26] (Supplementary Fig. 3b).

There has been a relative dearth of deep profiling of T cell subsets from BD patients. To address this, we first examined how BD impacted the composition of peripheral T cells in the scATAC-seq dataset and scRNA-seq dataset (Fig. 2c, Supplementary Fig. 5a, b). We saw similar trends in T cell subsets

**Fig. 1 High-throughput single-cell epigenomic and transcriptional profiling of Behcet's disease patients and health human peripheral blood cells.**
**a** Schematic highlighting design of single-cell multi-omics profiling of PBMCs from BD patients (scATAC: $n = 22$; scRNA: $n = 23$) and non-Behect's disease patients ($n = 8$) in this study. Cells were then split and profiled using scATAC-seq and scRNA-seq for each condition. **b** UMAP projections of complete scATAC-seq and scRNA-seq datasets between non-BD and BD patients with cells colored by unsupervised cell clusters. **c** Total number of six main immune cell types profiled passing quality control filtering for scATAC and scRNA-seq. **d** Aggregate accessibility profiles for scATAC-seq six main cell type at canonical cell-type marker genes. **e** UMAP projection of scATAC-seq peripheral blood profiles colored by *chromVAR* TF motif bias-corrected deviations for the indicated factors. **f** Row-normalized single-cell gene expression heatmap of six main immune cell-type marker genes. All data are aligned and annotated to hg38 reference genome.

composition between the multi-omics dataset, including decreased CD4 TFH, and CD4 TH2, as well as increased in CD8 Naïve, CD8 Treg, and DNT in patients with BD (Fig. 2c). DNT cells have been reported to be increased in BD patients, which is consistent with our finding[28]. TFs tightly control cell fate in immune cells and have been implicated in the pathogenesis of autoimmune diseases, such as BATF in arthritis and PU.1 in systemic lupus erythematosus[29–33]. We performed TF footprint analysis to further clarify cell-type-specific trans-regulatory elements in BD (Fig. 2d, Supplementary Data 3). We examined the regulatory role of the Treg TF AP-1 family, such as JUNB and FOSL2. JUNB and Fos-related antigen 2 (FOSL2) motif variabilities in our scATAC-seq were increased and upregulated in BD (Fig. 2d). This result suggests that the AP-1 family TFs might have higher accessibility in BD, providing insight into how the AP-1 family contributes to BD pathophysiology[34,35]. We therefore identified the DEGs and DARs between BD and non-BDs subjects in the scRNA-seq and scATAC-seq dataset (Fig. 2e, Supplementary Fig. 5c, d, Supplementary Fig. 6a, Supplementary Data 4). We also used muscat[36] to validate our DEGs result (Supplementary Fig. 7a, Supplementary Data 5). We next applied *CHIPseeker*[37] to find the nearest genes of the DARs and used the DARs to overlap with DEGs (Supplementary Fig. 6b). We observed that 14 genes were both upregulated in T cell subsets, including *DUSP2*[38], *JUNB*[39], *IRF1*[40], and *DDIT4*[39], suggesting T cells might be in proinflammatory state in BD patients. *CD5*[41], *CD69*[42], *NFKBIA*[43] were up-regulated in all the CD4 T cell subsets, suggesting CD4 T cell subsets were both highly activated in BD[42] (Supplementary Fig. 5c). In contrast, *CD7*[44], *IL2RG*[45], *IFITM1*[46], *IFITM2*[46] were up-regulated in all the CD8 T cell subsets (Supplementary Fig. 5d).

A Gene Ontology (GO) analysis of the DEGs showed that the cellular response to cytokine stimulus was enriched in most T cell lineages, suggesting the immune-activated states in BD patients (Fig. 2f, Supplementary Data 6). In T cells, the GO analysis showed that CD4 CTLs, CD8 Tex, and CD8 Treg were all enriched in the TCR signaling, and costimulation by the CD28 family pathway (Fig. 2f). The IL-4 signaling pathway was among the top enriched pathways in T helper cells, suggesting B cell-induced IL-4 mediated hyper-interplay with T cells in BD[47]. Th1 cells and Th17 cells were reported to take part in the BD pathogenesis[5,48,49] (Fig. 2f). We observed that Th1 cells were involved in the NF-kB signaling, TNF signaling pathway. Th17 cells were involved in the response to the interferon-gamma pathway with interferon signaling-related genes upregulated (*ISG15, IFITM1, IFITM2*) (Supplementary Fig. 7b). We also shed light on the significant number of rare T cell types, including MAIT and DNT cells. DNT cells were involved in the TCR signaling, T cell activation pathway, and HIF1 TF pathway (Fig. 2f). MAIT cells from BD patients showed increased enrichment for pathways associated with TNF signaling, NF-kB signaling, VEGFA-VEGFR2 signaling, and cellular response to cytokine stimulus (Fig. 2f). MAIT cells have been reported to contribute to the pathogenesis of other forms of vasculitis[8,50]. Overall, these analyses provide transcriptional and epigenomic

evidence that highly activated peripheral T cells may be associated with BD, consistent with previous single-cell reports[8].

**Multi-omic characterization of NK and B cell heterogeneity in BD blood.** Despite studies that show NK cells are involved in the dysregulated immune response in BD, the pathogenesis of NK cell subsets in BD still needs to be explored further. As demonstrated previously, peripheral NK cells were identified into three subsets based on the GAS and mark genes: early NKs (NK1; $NCAM1^{high}FCGR3A^{low}B3GAT1^{low}$), intermediate NKs (NK2, $NCAM1^{high}FCGR3A^{low}B3GAT1^{low}$) and late NKs (NK3, $NCAM1^{high}FCGR3A^{low}B3GAT1^{low}$)[17] (Fig. 3a, b, Supplementary Fig. 8a, b). While we did not notice a significant change in the percentage of total NK cells (Supplementary Fig. 8c, d), we noted significant transcriptional and epigenomic reconfiguration in all the NK subsets driven by up-regulated of several canonical NK cell activation genes (Supplementary Fig. 8e–g, Supplementary data 7), including *CD69*[51] as well as interferon-stimulated genes (ISGs) *IFITM2, IRF1*, and *ISG20* (Fig. 3c). NK1 also expressed higher cytotoxic effector molecule-encoding genes *GZMB, GZMM*, and *GZMH* (Fig. 3c).

To further elucidate the pathogenic pathways, we next examined DEGs of NK cell subsets for GO analysis (Fig. 3d). The top signaling pathways of NK1 in BD included the cytokine signaling pathway, regulation of T cell activation, IL-18 signaling pathway, CXCR4 pathway, costimulation by the CD28 family, and VEGFA-VEGFR2 signaling pathway (Fig. 3d). The pathway enrichment result implicates that NK1 cells in BD might be involved in the pro-angiogenic process. T-cell activation, CD28 co-stimulation, T-cell receptor signaling pathway, IL-6 signaling pathway, transcription regulation by RUNX1, and CD8 TCR downstream pathway were up-regulated in NK2, while regulation of the viral process, regulation of cell-cell adhesion, Th17 cell differentiation, cytokine-mediated signaling pathway, TNF signaling pathway, and FCGR3A-mediated phagocytosis were upregulated in NK3 (Fig. 3d). Both NK cell subsets in BD were enriched in the T cell activate and cytokine-related pathway with increased cytotoxic activity and up-regulated TNF pathway. The differences between non-BD suggest that NK cells in BD were in proinflammatory state[52]. The above findings suggested that NK-mediated immunity was activated and pro-inflammatory in BD patients.

We next interrogated changes in the B cells of BD patients, because B cells may play an eminent role in the pathogenesis[53] (Fig. 3e, Supplementary Fig. 9a). Embedding of B cells alone was manually identified into 4 subsets: naïve B cells (NB; $MS4A1^+TCL1A^+$), memory B cells (MB, $MS4A1^+TCL1A^+$), double negative B ells (DN2B; CD19+ITGAX+CR2−), and plasma B cells (PB, CD38+)[17] (Fig. 3f, Supplementary Fig. 9b). As observed in the transcriptional data, all the B cell subsets in BD highly expressed interferon-stimulated genes *IRF1*[40], *IFITM2*[46], and antigen processing and presentation-related molecules *HLA-DQB1*[54], and the cytokine *IL2RG*[55], as well as AP-1 family genes *JUNB*[56], MAPK signaling, and NF-kB signaling related genes *DUSP1*[38], *NFKBIA*[43] (Fig. 3g,

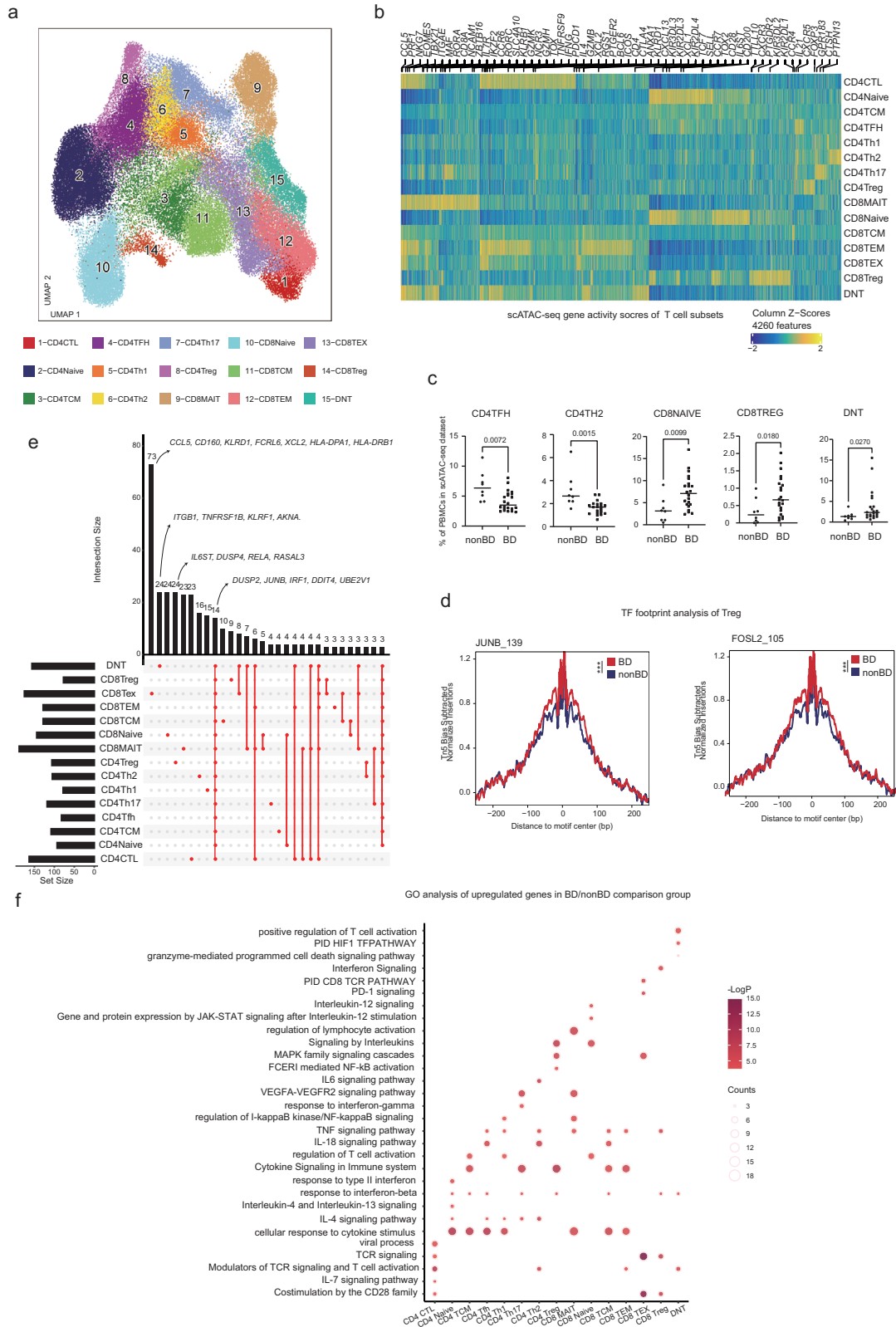

COMMUNICATIONS BIOLOGY | (2023)6:1048 | https://doi.org/10.1038/s42003-023-05420-x | www.nature.com/commsbio

Supplementary Fig. 9c–f, Supplementary Data 7). MB expressed higher *RELA*, *BACH1*, *SPIB*, *JUN* and *JUND*. DN2B showed higher expression on *ISG20*, *TNFRSF13C* and *BTG1* (Fig. 3g, Supplementary Data 7). By overlapping the DARs and DEGs, the activation marker *CD69* was highly expressed in DN2B[57], and the antigen-presenting and activation marker *CD83* was highly expressed in MB and NB, suggesting a highly activated state of B cells in BD patients[58] (Supplementary Fig. 9e, f).

Next, we performed GO analysis on upregulated genes in BD patients (Fig. 3h). The GO results showed that DN2B in BD patients were enriched in positive regulation of T cell activation, regulation of IL-2 production, activator protein 1 (AP-1)

**Fig. 2 scATAC-seq and scRNA-seq analysis of the changes in T cell subsets in BD. a** UMAP projections of T cell subsets of scATAC-seq dataset.
**b** Heatmap visualization of log-normalized gene activity scores of subpopulation-specific genes in T cell subsets. Selected genes are indicated.
**c** Differences in the proportions of CD4Tfh, CD4Th2, CD8 Navie, CD8 Treg and DNT cells among non-BD ($n = 8$) and BD groups ($n = 22$). The p values
were calculated using two-sided Wilcoxon rank-sum test. The horizontal lines denote median. **d** Comparison of aggregate TF footprints for JUNB and
FOSL2 in CD4Treg cells from non-BD and BD. The p value of the TF footprint was compared by two-sided Wilcoxon rank-sum test. **e** UpSet plot showing
the integrated comparative analysis of upregulated DEGs in T cells between non-BD and BD groups. Upregulated DEGs: upregulated in BD, downregulated
in non-BD. The count shows the number of DEGs. **f** Representative GO terms and KEGG pathways enriched in the nearest genes of upregulated DARs of T
cell subsets in the BD/non-BD comparison group. The p values were calculated by hypergeometric test. All data are aligned and annotated to hg38
reference genome.

pathway, PD-1 signaling, and costimulation by the CD28
family[59], while PB were enriched more in IL-1 signaling FCERI
mediated NF-kB activation, Dectin-1 signaling and cellular
response to hypoxia (Fig. 3h). Moreover, GO analysis suggested
that MB from BD patients were enriched in the AP-1 pathway,
IL-18 signaling, MAPK signaling, signaling by interleukins, and
response to the interferon-beta pathway (Fig. 3h). As for NB,
Antigen processing and presentation, major histocompatibility
complex (MHC) protein complex assembly, regulation of T cell
activation, response to the virus, and TCR signaling pathway were
up-regulated (Fig. 3h). Taken together, these findings indicated
common and distinct functions of B cell subsets among non-BDs
and BD patients and suggested an enhanced humoral immunity is
developed in BD patients.

**Multi-omic characterization of myeloid cells heterogeneity in
BD blood.** To increase cell-level resolution and dissect myeloid
cells, we first clustered DCs and identified 3 DC subtypes: type 1
classical dendritic cells (cDC1; HLA+CLEC9A+), type 2 classical
dendritic cells (cDC2; HLA+ITGAX+), and plasmacytoid DCs
(pDCs; HLA+IRF8+CLEC4C+)[17,20] (Fig. 4a, Supplementary
Fig. 10a–c). We first examined how BD impacted the composition
of peripheral DCs in two datasets (Supplementary Fig. 10D,
S10E). We saw cDC2 significantly increased in the scRNA-seq
dataset (Supplementary Fig. 10d). Next, we analyzed the differ-
ence between BD and non-BD in transcription and epigenomic
profiling (Fig. 4b–e, Supplementary Fig. 10f, g, Supplementary
Data 5). Notably, cDC2 in BD patients showed significantly
higher chromatin accessibility on the IL-1B locus and CD83
locus, suggesting increased cytokine secretion and enhanced
antigen presentation in DCs (Fig. 4b, c). To further access the TF
that drove differences in cDC2, we conducted a TF footprint
analysis compared to BD and non-BD in cDCs (Fig. 4d, Sup-
plementary Data 1). The results showed that NFKB family TF
(RELB, NFKB1) showed high accessibility in BD patients.
Consistent with our epigenomic dataset, the cytokines and
chemokines CCL3L1, CCL3, CCL4L2, IL1B, and ISGs IFITM3,
IRF1, IFITM1, ISG20, as well as activation marker CD83 and NF-
kB signaling related genes (RELB) were up-regulated in cDC2
(Fig. 4e). Interestingly, HIF1A was also up-regulated in cDC2,
suggesting epigenetic reprogramming and metabolism changes in
cDC2[60] (Fig. 4e). Furthermore, cDC2 with high expression of
NFKBIA, JUNB, and MAP3K8 showed GO enrichment in TNF
signaling, regulation of T cell activation, interferon signaling, IL-
18 signaling, and AP-1 pathway based on GO analysis (Fig. 4e,
f)[61,62]. The GO analysis showed that the DEGs of cDC1 in BD
patients were enriched in the Dectin-1 signaling, gluconeogenesis,
interleukin-1 signaling, MAPK1/MAPK3 signaling, and signaling
by interleukins pathways. While the DEGs of pDC were enriched
in TCR signaling, glycolysis, and gluconeogenesis, Fc epsilon
receptor signaling (Fig. 4e, f). Our analysis of DCs showed that
DC subsets in BD patients varied greatly compared to those in
non-BD, suggesting enhanced DC function and further directed
T cells differentiation in BD patients[63].

It has been reported that monocytes play vital roles in BD[4],
however, the epigenomic changes of monocytes in BD have not
been well characterized. To address this, we sub-grouped
monocytes into 4 subclusters, which consisted of classical
monocytes based on the GAS and DEGs between each cluster.
The classical monocytes (CM) were identified based on the
expression of CD14+FCGR3A-, while the non-classical monocytes
(NCM) expressed more FCGR3A (Fig. 4g–i, Supplementary
Fig. 11a, b, Supplementary Data 8). Notably, cell subsets
proportions were altered in BD patients. BD patients had a large
proportion of monocytes that highly expressed cytokines and
chemokines IL1B, CXCL8, CCL4L2, and CCL3 with
CD14+S100A12+FCGR3A-, which we identified as activated
classical monocytes (ActCM)[43]. We also identified previously
reported C1Q+ monocytes with a high expression on ISG
IFITM2, and pro-inflammatory genes TNF, IL1B, CXCL8,
CCL4L2, CCL3, as well as FCGR3A, RHOC, which we identified
as activated non-classical monocytes (ActNCM)[4,43,64] (Fig. 4g–i).
Comparing the relative cell proportions in BD and non-BD
groups, we observed a similar trend to what was previously
reported in single-cell profiling[4], a significant expansion in
ActCM and ActNCM, and a notable decrease in NCM in the
scATAC-seq dataset, while the proportion of CM was comparable
between BD and non-BD groups (Fig. 4j, Supplementary Fig. 11c,
Supplementary Data 2). Although we did not notice significant
changes in ActNCM in the scRNA-seq dataset, we still observed a
significant increase in ActCM (Supplementary Fig. 11c).
Furthermore, within these subpopulations, we conducted GO
analysis (Fig. 4k). Both ActCM and ActNCM were enriched in
cytokine signaling in the immune system pathway (Fig. 4k,
Supplementary Data 6). GO enrichment analysis of upregulated
genes in ActCM highlighted strong signatures for IL-18 signaling,
NF-kB signaling, regulation of myeloid cell differentiation, TNF
signaling, Toll-like receptor signaling, and positive regulation of
cell death pathway. Relative to other sub-population, the GO
analysis showed ActNCM were enriched for signaling by
interleukins, positive regulation of cytokine production, regula-
tion of T cell activation, regulation of cell-cell adhesion, the
VEGFA-VEGFR2 signaling pathway, and regulation of tumor
necrosis factor production (Fig. 4k).
We next performed TF deviation analysis on monocyte subsets,
we noticed monocyte subsets exhibited deviated variations in
different TF family members from homeostasis to the activated
effector state (Supplementary Fig. 11d). For example, both
ActCM and ActNCM were associated with high levels of activity
of TF involved in NF-kB signaling signatures and myeloid
differentiation, including REL, RELA, NFKB1, and NFKB2[64,65]
(Supplementary Fig. 11d). Interestingly, ActCM had increased
enrichment in TFs that represent activation and maturation
stages, such as AP-1 family members, FOS, and JUN[56,64,66]. In
addition, CM was also enriched in the activation of AP-1 family
TFs (Fig. 4l). Meanwhile, ActNCM showed high activity of TFs
involved in haematopoetic commitment and survival of mono-
cytes, including NR4A1 and NR4A2[64,67] (Fig. 4l, Supplementary

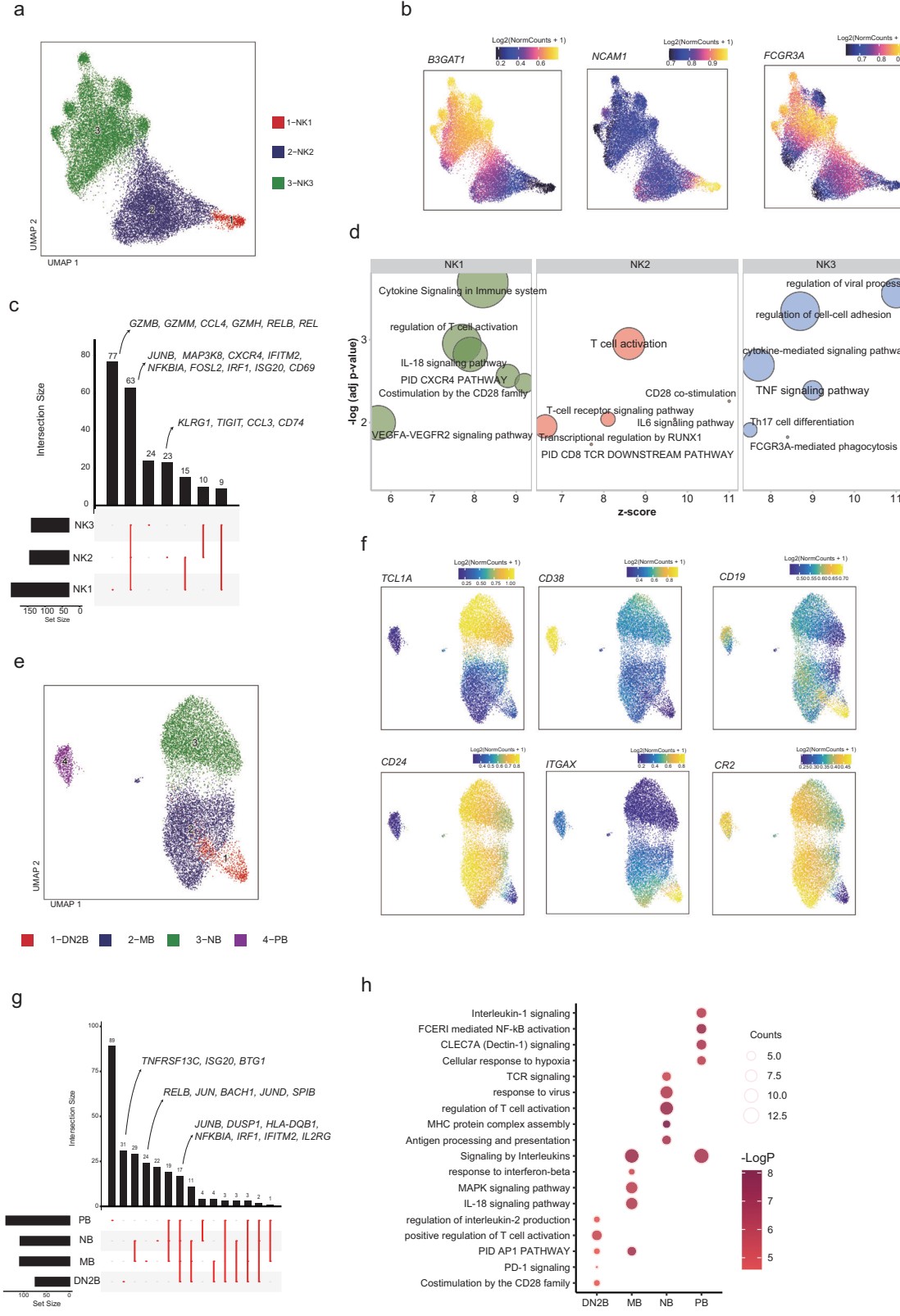

Fig. 11d, e, Supplementary Data 1). Collectively, myeloid subsets maintained chromatin reprogramming and transcription changes that promote a rapid inflammatory response in BD.

**Multi-omic integration mapping enables cellular annotation and analysis**. We reasoned that integrating data from scATAC-seq and scRNA-seq datasets may enable the determination of gene-regulatory networks (GRNs) by data integration, facilitating the interpretation of the key regulatory processes underlying the pathogenesis of BD. Based on the sub-clustering and manual cellular annotation we described above, we identified 29 immune cell subsets in 20,000 cells (Fig. 5a). We also automated annotated cell types using *clustifyr*[68] package based on Hao et al.[23] to

**Fig. 3 scATAC-seq and scRNA-seq analysis of the changes in NK cell and B cell subsets in BD. a** UMAP projections of NK cell subsets of scATAC-seq dataset. **b** UMAP projection colored by gene activity scores for the annotated lineage-defining genes of scATAC-seq dataset. The minimum and maximum gene activity scores are shown in each panel. **c** UpSet plot showing the integrated comparative analysis of upregulated DEGs in NK cells between non-BD and BD groups. Upregulated DEGs: upregulated in BD, downregulated in non-BD. The count showing the number of DEGs. **d** Representative GO terms and KEGG pathways enriched in the upregulated DEGs of NK cell subsets in the BD/non-BD comparison group. The *p* values were calculated by hypergeometric test. **e** UMAP projections of B cell subsets of scATAC-seq dataset. **f** UMAP projection colored by gene activity scores for the annotated lineage-defining genes of scATAC-seq dataset. The minimum and maximum gene activity scores are shown in each panel. **g** UpSet plot showing the integrated comparative analysis of upregulated DEGs in NK cells between non-BD and BD groups. **h** Representative GO terms and KEGG pathways enriched in the upregulated DEGs of B cell subsets in the BD/non-BD comparison group. The *p* values were calculated by hypergeometric test. All data are aligned and annotated to hg38 reference genome.

validate our manual annotation based on known marker genes (Supplementary Fig. 12a, b). Next, we interrogated the BD and non-BD datasets from epigenomic and transcriptomic data and utilized the current frameworks supporting the integration of scATAC-seq and scRNA-seq data, relying on identifying mutual nearest neighbors cells – cells, which represent shared biological states in a common lower-dimensional space - to then find representative cells from one dataset in the other (Supplementary Fig. 13a–c). The whole process was parallelized and separately aligned using the *ArchR*[24] and *Seurat*[69,70] pipelines by separating cells into smaller groups. This procedure enabled us to accurately integrate the transcriptomic data from the scRNA-seq dataset with the chromatin accessibility data from the scATAC-seq dataset by mapping the GAS and gene expression to generate an integration matrix. As expected, the GAS and gene expression were matched, which allowed us to distinguish the 29 immune cell types (Fig. 5b, Supplementary Fig. 13d).

Dissecting the molecular mechanisms behind autoimmune disease complex phenotypes identified by genome-wide association studies (GWAS) requires pinpointing disease-relevant cell types. However, nearly 90% of causal genetic variants lie in noncoding regions[31]. In addition, much work has shown that the resolution of intersecting GWAS signals with bulk data is impeded by cell-type heterogeneity[71–73]. We wondered if we could use our single-cell data to better dissect the cell-type-specific effects of genetic variations underlying complex human autoimmune disease traits. To address these issues, we adopted the *g-chromVAR*[74] method on our global single-cell chromatin data to identify trait-cell type associations in a peripheral immune cell subsets-dependent manner (Fig. 5c).

We used a publicly available database for autoimmune and non-immune disorders from a previous study[31] and calculated the enrichment of disease-related SNPs in 29 peripheral immune cell types using the *g-chromVAR*[31,74]. The majority of the autoimmune associations were strongly enriched for a corresponding trait association. For example, BD was significantly enriched in CD4 CTL, CD8TEM, DNT, and CD8TCM cells[75,76], while type 1 diabetes was most strongly enriched in Th17 cells (Fig. 5c)[77]. Within the open chromatin region of our broad cell subsets, *g-chromVAR* enrichment revealed significant T cell subsets, reinforcing the previous study[22,75,76]. T cells with cytotoxicity have been reported to contribute to BD pathogenesis in both skin and circulation[8]. Additional trends of T cells enrichment are also observed here for type 1 diabetes and asthma, although no statistically significant. Most of these non-autoimmune disease GWAS were not apparent in the immune cell peaks, demonstrating GWAS enrichment was consistent with our expectations. Although not the focus of our current study, we observed that our generated PBMC chromatin data could provide cellular-specific enrichment of human autoimmune disease heritability.

We compiled a list of 66 index SNPs from Farh et al.[31] representing GWAS hits for BD. We then identified all the SNPs

in scATAC peaks and focused on their nearest genes. Furthermore, we calculated peak-to-gene connections for these gene locus using ArchR (Fig. 5d, Supplementary Fig. 13e–g). We noticed the rs201985743[78], that confers the risk of BD was in the *KLRC4-KLRK1* enhancer region, which was opened in CD8 T cell subsets and NK cells subsets[79]. This enhancer was highly accessible in NK and CD8 T subsets, but not in T cells, B cells or monocytes, demonstrating NK and CD8 T specificity. In *KLRC4-KLRK1* enhancer region showed statistically increasingly strong peak-to-gene linkages in non-BDs compared to BD patients (Fig. 5d, Supplementary Fig. 13g). There are no differences in the expression of *KLRC4-KLRK1* in NK and T cell subsets in the scRNA-seq dataset (Supplementary Data 4, 7, 9). However, we did not notice other SNP loci showed stronger predictive linkages between BD and non-BDs. The low peak accessibility and gene expression in the BD state illustrates chromatin dynamic regulation in the *KLRC4-KLRK1* locus. Since Killer cell lectin-like receptor subfamily (KLRC) regulates NK and CD8T function[79–81], it is possible that this SNP contribute to the pathogenesis of BD by dysregulating *KLRC4-KLRK1* in NK and CD8 T function. Therefore, our multi-omic integration analysis could provide predictive disease mechanisms that involve alterations in BD chromatin and gene regulatory regions.

**Multi-omic integration analysis identifies candidate TF regulators of DORC activity.** The modulation of gene expression by changes in chromatin accessibility is crucial to understating the pathogenesis of autoimmune disease. To address this, we have created well-integrated multi-omic data and used FigR[82] to deduce key transcriptional regulatory networks that are required for BD pathogenesis. We first used our scATAC-seq and scRNA-seq data for *ChromVAR* enrichment of TF motifs among pre-determined cis-regulatory elements, as well as the correlation of TF deviation score with the overall chromatin accessibility level for gene activity scores of TF genes, to infer likely positive TF regulators[24] (Fig. 6a). We noted that in our dataset, ETS family TFs such as SPIB and SPI1 (also known as PU.1) ranked as the top two TFs related to IFN stimulation and MHC class II gene expression[83,84] (Fig. 6a). In addition, AP-1 family TFs JUNB and FOS, and NF-kB family TFs NKFB1 and RELB, also showed high activity in our multi-omic dataset.

To connect distal cis-regulatory elements to genes and infer a GRN, we first utilized our scATAC-seq and scRNA-seq data. We next defined domains of regulatory chromatin (DORCs). We used a computational approach ($n = 150,000$ cells per assay), FigR framework, to determine DORC within a fixed window (100 kb) around the transcription start site of each gene (Fig. 6b–d). In this way, we identified a total of 23,627 unique cis-regulatory associations genome-wide, showing significant chromatin accessibility peaks with gene expression (permutation $p <= 0.05$). We defined 202 regions with an exceptionally large (>7) number of significant peak-gene associations as DORCs, identified as those exceeding an inflection point (Fig. 6b,

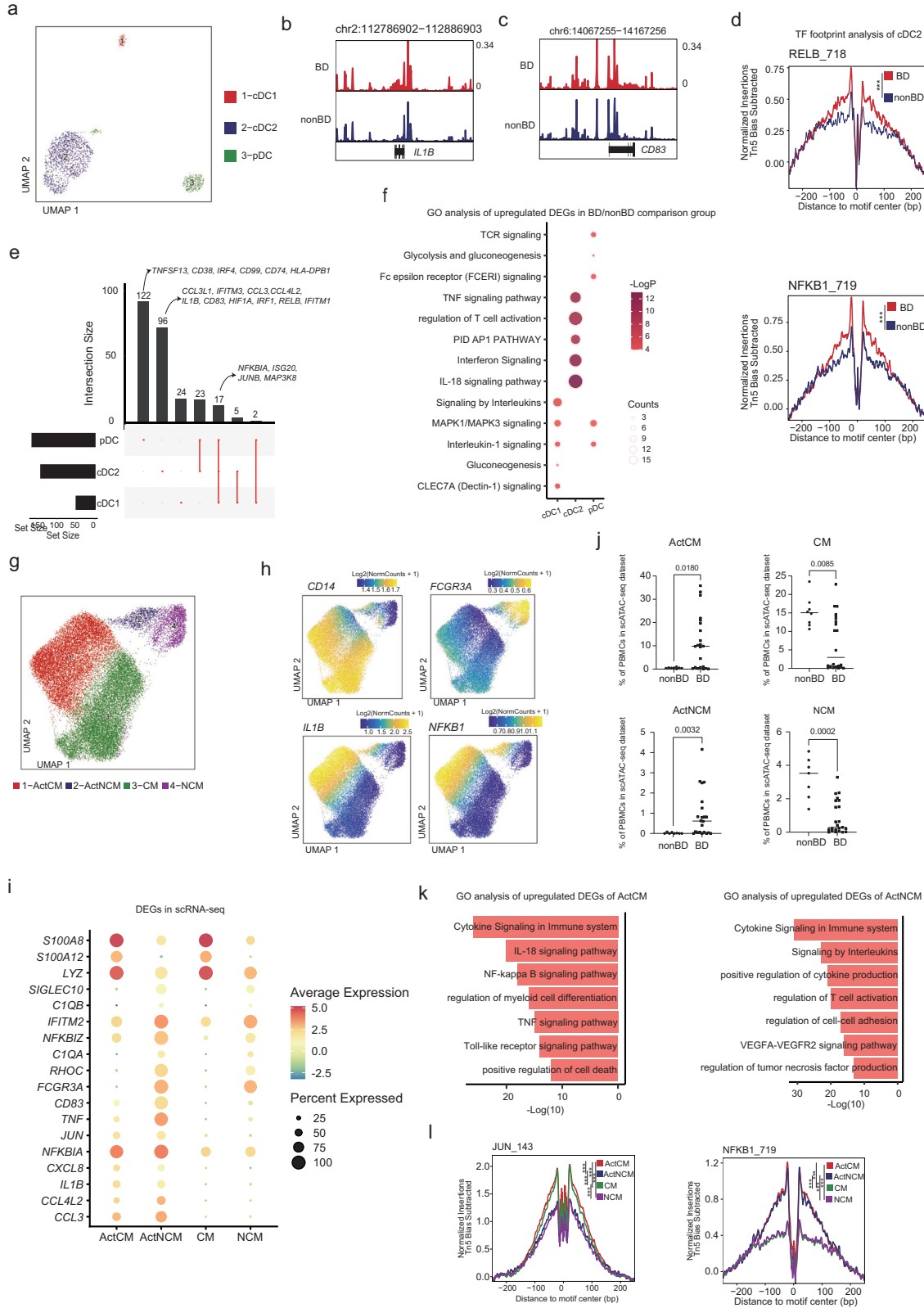

Supplementary Data 10). It has previously been reported that DORCs are highly cell-type specific[85]. We noted that the top genes were *SEMA7A*, *PIK3R2*, *HLA-DQB1*, and *IL10*, which included many well-known mediators of immunological response associated with innate and adaptive immune response pathways (Supplementary Fig. 14a). For example, the top GO enrichment pathways were regulation of T cell activation, B cell receptor

signaling pathway, costimulation by the CD28 family, cytokine signaling, and TCR signaling pathway. Notably, we also observed that the VEGF signaling pathway and positive regulation of leukocyte adhesion to vascular endothelial cells were also on top of the GO analysis, suggesting the possible pathogenesis of vasculitis in BD patients (Supplementary Fig. 14a). Although the top-ranked gene was SEMA7A, we did not observe a significant

**Fig. 4 scATAC-seq and scRNA-seq analysis of the changes in DC and Monocytes subsets in BD. a** UMAP projections of DC cell subsets of scATAC-seq dataset. **b** Genome browser tracks showing single-cell chromatin accessibility of cDC2 cells in the *IL1B* loci. **c** Genome browser tracks show single-cell chromatin accessibility of cDC2 cells in the *CD83* loci. **d** Comparison of aggregate TF footprints for RELB and NFKB1 in cDC2 cells from non-BD and BD. The *p* value of the TF footprint was compared by two-sided Wilcoxon rank-sum test. **e** UpSet plot showing the integrated comparative analysis of upregulated DEGs in DC subsets between non-BD and BD groups. Upregulated DEGs: upregulated in BD, downregulated in non-BD. The count showing the number of DEGs. **f** Representative GO terms and KEGG pathways enriched in the upregulated DEGs of DC cell subsets in the BD/non-BD comparison group. The *p* values were calculated by hypergeometric test. **g** UMAP projections of monocyte subsets of scATAC-seq dataset. **h** UMAP projection colored by gene activity scores for the annotated lineage-defining genes of scATAC-seq dataset. The minimum and maximum gene activity scores are shown in each panel. **i** Dot plots of gene expression of the marker genes of monocyte subsets in scRNA-seq dataset. The dot size indicates the percentage of the cells in each cluster in which the gene of interest. The standardized gene expression level was indicated by color intensity. **j** Differences in the proportions of monocyte subsets among non-BD ($n = 8$) and BD groups ($n = 22$). The *p* values were calculated using two-sided Wilcoxon rank-sum test. The horizontal lines denote median. **k** Representative GO terms and KEGG pathways enriched in the marker genes of the ActCM and ActNCM. The *p* values were calculated by hypergeometric test. **l** TF footprints with motifs in the indicated scATAC-seq monocyte subsets. The *p* value of the TF footprint was compared by one-way ANOVA. All data are aligned and annotated to hg38 reference genome.

---

difference between BD and non-BD (Supplementary Fig. 14B). However, we noted that HLA-DQB1 was highly expressed in the BD group compared to non-BD (Supplementary Fig. 14B). Next, we calculated the correlation of a given DORC gene to TF expression and further queried putative TF regulators for a given DORC (Fig. 6c, d). We identified known activators of *HLA-DQB1*, including the CCAAT/enhancer-binding protein (CEBP) family TF members: CEBPA, NF-kB family TF members: NFKB2, and RELB (Fig. 6d)[86]. Importantly, RELB has been reported to be a key activator in DC maturation[87].

Based on the above analysis, we inferred that NF-kB family TFs, AP-1 family TFs, and ETS family TFs might work as key pro-inflammatory factors in BD. Leveraging DORCs and the TFs that regulate them, we constructed a GRN that underlies the peripheral blood immunity of BD, relating each TF to each DORC (Fig. 6e, Supplementary Data 11). Consistent with previous reports, AP-1 family TFs are central components of regulatory factors for IL-10 expression[88]. Using a computational method, we demonstrated the ability to identify DORCs and GRN to determine disease-specific chromatin accessibility profiles relevant to autoimmunity.

## Discussion

BD is a complex immunogenic and systemic disease for which the pathogenic cell type and pathway have not yet been identified, leading to a lack of targeted treatment for this disease[1,2]. Studies of autoimmunity in BD pathogenesis have been particularly limited by mouse models that do not have equivalent clinical phenotypes[89]. Leveraging scATAC-seq and scRNA-seq to accurately map chromatin accessibility and gene transcription and predict underlying TF regulators and GRN within BD blood samples, provides a unique and physiological approach to elucidate the autoimmunity landscape in this complex disease.

Here, we analyzed the chromatin and transcript landscape underlying human peripheral blood BD heterogeneity by delineating the repertoire of accessible cis-elements and genes in a multi-omic manner. Through our single-cell atlas of peripheral blood and multi-model analysis approach, our work revealed: (1) single-cell epigenomic and transcriptomic profiles of peripheral blood in BD patients; (2) widespread activation of peripheral autoimmunity profiles in BD patients; (3) putative TF activators that drive the changes chromatin accessibility in BD patients; and (4) potential GRN of BD-associated regulatory interactions within putative TF regulators. Importantly, we described the single-cell regulatory peripheral immunity atlas of BD patients, which provides insights into the chromatin level of blood autoimmunity landscape of BD.

Previous single-cell studies on BD included scRNA-seq and scTCR-seq but did not involve scATAC-seq[4,8]. Both single-cell

studies that described the peripheral immunity of the BD have reported enhanced interferon signaling, which also was noted in our data. C1q+ monocytes have been reported to expand in the blood of BD patients by activated IFN-signaling[4]. CD4+ Treg cells have also been reported to increase in the affected skin tissue of BD patients[8]. Although we did not notice a significant expansion of CD4+Treg cells in our data, we noticed CD8+Treg cells and Act NCM have significantly higher proportions compared to non-BD. Moreover, we did not notice the frequency of cytotoxicity CD8+ T cell expanded. However, our computational analysis predicted that the GWAS enrichment of open chromatin analysis linked the probable causal BD variants to specific blood cytotoxic CD8$^+$ T cell subsets. It has been previously reported that circulating CD8$^+$ T cell might share a clonal origin with skin-filtrating CD8 T cells and acquire tissue-residential features leading skin lesions[8]. CD8+ Treg cells expressing inhibitory killer cell immunoglobulin-like receptors (KIRs) have been described as increased in the blood and inflamed tissues of patients with a variety of autoimmune disorders and reported to elevate in COVID-19 patients[27]. The elevated levels of CD8+ Tregs were also related to COVID-19 vasculitis[27]. Although the expansion of CD8+Treg in BD may act as a negative feedback mechanism to ameliorate inflammation in peripheral blood, this indicates that CD8+Tregs represent an important element in peripheral tolerance and BD pathophysiology. In total, we provided a more comprehensive blood immune cell landscape of BD.

Our analysis enabled identification of BD-associated SNPs that lie in the regulatory regions of cytotoxicity subsets. Furthermore, we observed that the predictive peak-to-gene linkages near the rs201985743 loci within the *KLRC4-KLRK1* region showed significantly stronger linkages in non-BDs compared to BD. Killer cell lectin-like receptor subfamily member 4 (KLRC4, belonging to the NKG2 receptor family known to play an important role in regulating NK and T cell functions[81,90], has previously been linked to BD[91]. The interactions between peaks and genes may indicate physical interaction of the regulatory region affecting its target genes in NK and CD8$^+$ T[92–94]. The differential interaction between the *KLRC4-KLRK1* locus in NK cells and CD8$^+$ T cells between non-BD and BD groups may suggest gene expression effects of the causal variant[93]. However, the molecular cause of NK and CD8 T dysregulation is still unknown. This approach can allow us to predict gene and cellular targets in BD and nominate the most disease-relevant cell types and meriting functional validation.

Importantly, we also generalized multi-model datasets assaying chromatin accessibility and gene expression to infer possible TF activators and their potential GRN that might drive disease-associated phenotypes. This method has been utilized in anticipating in cells from diverse stimuli, cellular differentiation, and

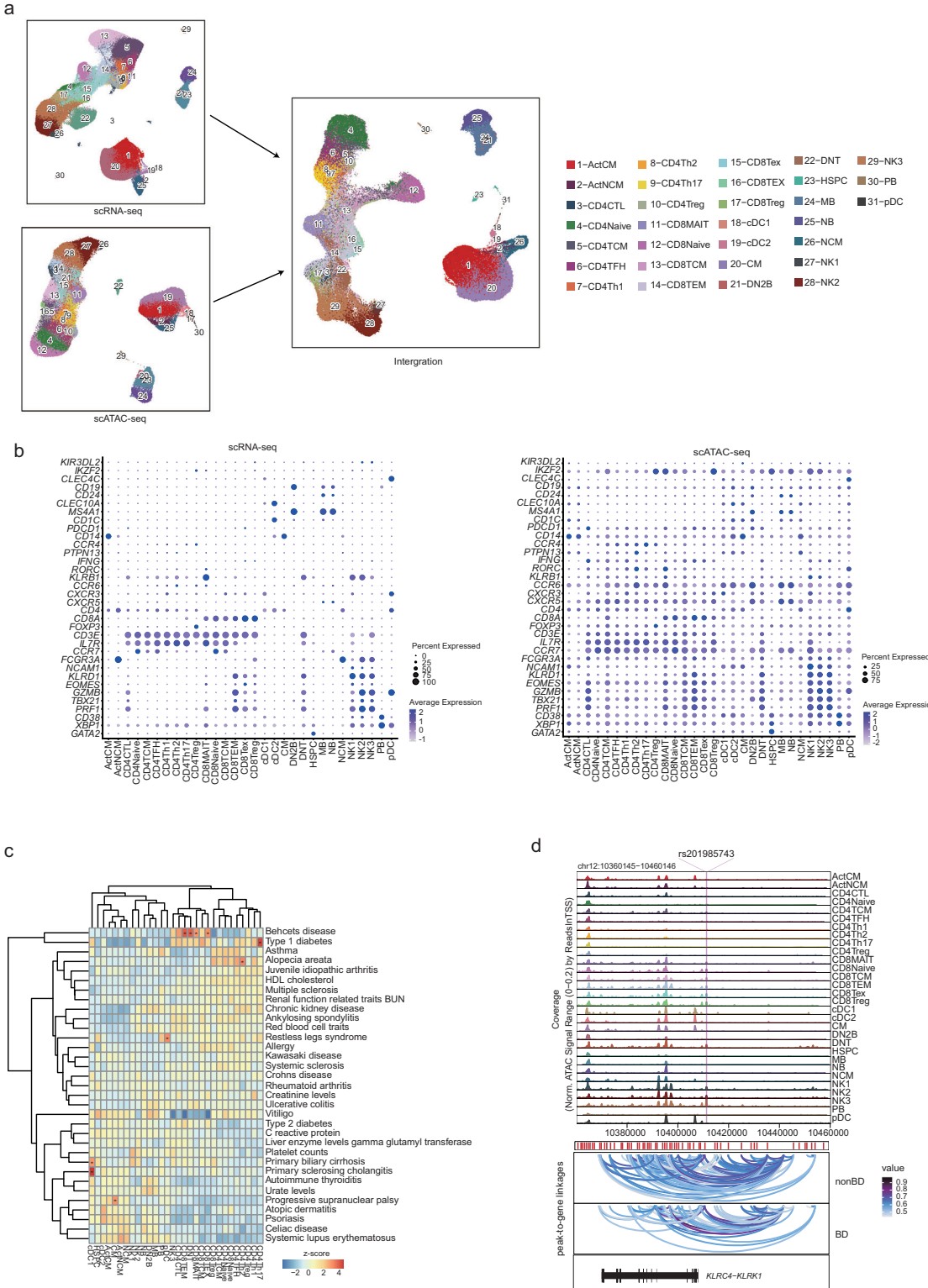

even in disease states, providing additional insights into the regulatory mechanisms underlying gene priming. PU.1, AP-1, and NF-kB TFs were predicted and nominated as putative TFs that induced peripheral inflammation of BD, These TFs have been reported to be essential transcription factors regulating multiple inflammatory pathways and contributing to autoimmunity[31].

While the causative molecular mechanisms of BD remain elusive, our work offers insights into elucidating the nature of gene regulation in BD, especially regarding the comprehensive single-cell multi-omic landscape of epigenetic and transcriptional patterns. The dataset presented here is a valuable resource for understanding the regulatory relationship in human auto-immunity, and our analysis makes discoveries in chromatin

**Fig. 5 Integrative multiomic analysis in human PBMCs of BD patients and non-BD individuals. a** Schematic for multiomic integration strategy for processing the scRNA-seq and scATAC-seq dataset. Following integration and label transfer. Dots represent individual cells, and colors indicate immune cell types (labeled on the right). **b** Dot plots of gene activity scores (left) and gene expression (right) of the marker genes in scATAC-seq and scRNA-seq dataset. The dot size indicates the percentage of the cells in each cluster in which the gene of interest. The standardized gene activity score level (left) and gene expression level (right) were indicated by color intensity. **c** Enrichment for autoimmune disease-associated SNPs performed by *g-ChromVAR*. Color indicates enrichment score. The adjusted *p* values were calculated using Mann–Whitney *U* test and Benjamini–Hochberg test. **d** Cis-regulatory architecture at the following GWAS loci and cell types in PBMCs: *KLRC4-KLRK1*. Only connections originating in the loci with peak-to-gene accessibility above 0.4 are shown. All data are aligned and annotated to hg38 reference genome.

accessibility and gene expression using single-cell multi-omic data. Finally, we represent peripheral immune responses in BD and enhance our understanding of epigenomic pathological immune responses in BD.

## Methods

**Human subjects**. The study was approved by the Ethics Committee of Zhongshan Ophthalmic Center (Guangzhou, China, 2019KYPJ114), and followed the relevant ethical regulations for human research participants according to the Declaration of Helsinki. Written informed consent was obtained from all participating individuals, who were recruited from Zhongshan Ophthalmic Center. Exclusion criteria for the study included comorbid conditions such as cancer, immunocompromising disorders, hypertension, diabetes, and steroid use. The non-BD group consisted of eight individuals, 4 men and 4 women, with an average age of 46.8 years (Supplementary Data 1). The BD patient cohort (Supplementary Data 1) comprised 13 men and 10 women, with an average age of 33.3 years, diagnosed based on the revised diagnostic criteria established by the 2013 International Criteria for BD[95].

**Cell isolation**. To isolate peripheral blood mononuclear cells (PBMCs), peripheral venous blood samples were taken from non-BD individuals or BD patients and treated with a Ficoll-Hypaque density solution and heparin. The mixture was then centrifuged for 30 min. The single-cell suspensions were stained with Trypan blue to assess viability and quantity. Only those samples with cell viability over 90% were selected for subsequent experiments. For each sample that contained over $1 \times 10^7$ viable cells, a portion of PBMCs was extracted for scRNA-seq analysis while reserving another fraction for scATAC-seq assays.

**scATAC-seq processing**. The single-cell nuclei were isolated, washed and counted following the manufacturer's protocols. To obtain the desired final concentration based on the number of cells, an appropriate volume of Diluted Nuclei Buffer (10x Genomics; PN-2000153) was utilized to resuspend nuclei. The nuclei concentration was then determined using a Countess II FL Automated Cell Counter. Isolated nuclei were promptly used to create 10x single-cell ATAC libraries at Berry Genomics Co., Ltd. (Beijing, China). Each library was uniquely barcoded and quantified by RT-qPCR before being loaded onto an Illumina Novaseq 6000 with a loading concentration of 3.5 pmol/L in pair-end mode. Sequencing was performed until 90% saturation or an average of 30,000 unique reads per cell were acquired. The protocols for sample processing, library preparation, and instrument and sequencing settings on the 10x Chromium were adhered to as described in https://support.10xgenomics.com/single-cell-atac. Raw sequencing data were demultiplexed to fastq format using 'cellranger-atac-mkfastq' (10x Genomics, v.1.0.0). Subsequently, the scATAC-seq data reads were aligned to the GRCh38 (hg38) reference genome and quantified using the 'cellranger-atac count' function (10x Genomics, v.1.0.0).

**scATAC-seq quality control**. To generate Arrow files, *ArchR* v0.9.5[24] was utilized to analyze the accessible read fragments of each sample, following the default settings. To ensure high signal and sequencing quality, cells with less than 2500 unique fragments and TSS enrichments below 9 were filtered out. Doublets were inferred and eliminated using *ArchR* with default parameter, while cells that mapped to blacklist regions based on the ENCODE project reference were also excluded.

**scATAC-seq dimensionality reduction and clustering**. To reduce dimensionality, we employed a layered approach to reduce dimensionality using techniques such as latent semantic indexing (LSI) and singular value decomposition (SVD). The single-cell accessibility profiles were clustered utilizing *Seurat*'s shared nearest neighbor (SNN) 21 graph clustering with a default resolution of 0.8. We then reclustered using 'FindClusters' at a resolution of 0.8 to improve the identification of small clusters. Finally, we utilized uniform manifold approximation and projection (UMAP) to visualize all data in two-dimensional space. We did not detect potential batch effects in our dataset. Therefore, no batch correction method was applied in our further analysis.

**scATAC–seq gene activity scores**. To calculate gene activity scores, we utilized *ArchR*[17] v.0.9.5 with default parameters, correlating accessibility at the gene body, promoter, and distal regulatory elements with gene expression. The *MAGIC*[96] imputed weight method was then applied to the resulting gene activity scores to reduce noise due to the sparsity of scATAC-seq data.

**scATAC–seq pseudobulk replicate generation and peak calling**. To enable differential comparisons of clusters, cell types, and clinical states, we created non-overlapping pseudobulk replicates from groups of cells using the 'addGroupCoverages' function, by varying the arguments. These pseudobulk replicates were then employed to create the peak matrix using 'addReproduciblePeakSet' function. To identify peaks, we utilized *MACS2*[97] tool for peak calling. Finally, we utilized the pseudobulk peak set for downstream analysis.

**scATAC–seq genomic regions annotation**. In the differential analysis, we utilized the *ChIPseeker*[37] package's "annotatePeak" function with default arguments to annotate the nearest genes in the peak region.

**scATAC motif enrichment and motif deviation analysis**. Motif enrichment and motif deviation analyses were conducted on the pseudobulk peak set. We employed the Catalog of Inferred Sequence Binding Preferences (CIS-BP) motif (from *ChromVAR*)[24,98], *JASPAR2020* motif[99] and *HOMER*[100] to perform peak annotation. Furthermore, we utilized the *ArchR* implementation to calculate the *chromVAR* deviation scores for these motifs.

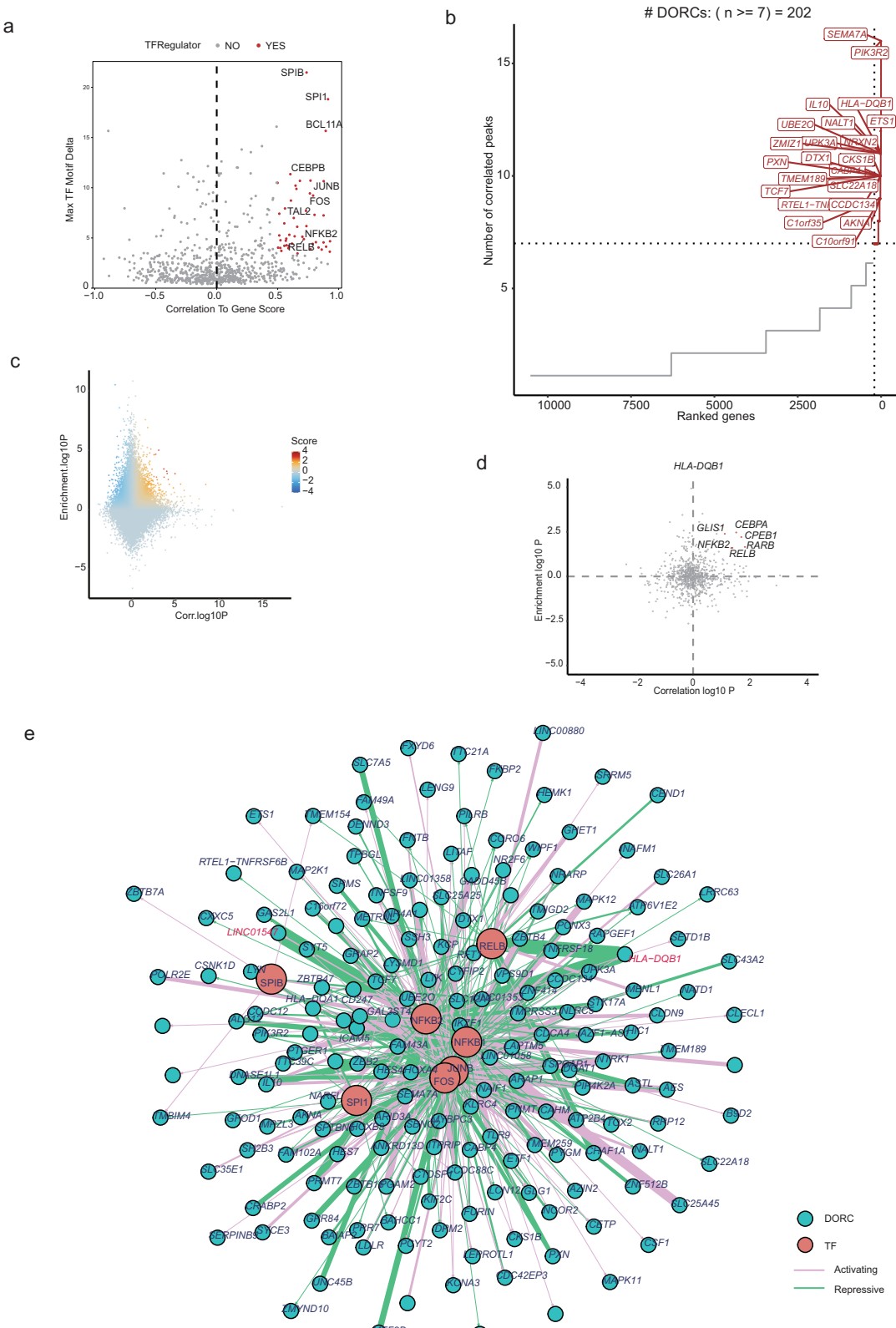

**Fig. 6 Identification TF regulators and gene regulatory network in the peripheral blood of BD patients and non-BD individuals. a** Volcano plot of positive TF regulators using gene expression of the TF and inferred gene activity score. **b** Top hits based on the number of significant gene-peak correlation across all cell types. Genes with >7 linkages are defined as domains of regulatory chromatin (DORCs). **c** Scatterplot showing all DOC-to-TF associations, colored by the signed regulation score. **d.** Candidate TF regulators of *HLA-DQB1*. Highlighted points are TFs with abs(regulation score) >=1 (−log10 scale), with all other TFs shown in gray. **e** TF-DORC network visualization for candidate TF positive regulators-implicated DORCs (green nodes) and their associated TFs (red nodes). Edges are scaled and colored by the signed regulation score. Highlighted points in red are TFs with abs(regulation score) >=1 (−log10 scale). All data are aligned and annotated to hg38 reference genome.

**scATAC–seq TF Foot-print analysis**. To conduct motif footprint analysis, we measured Tn5 insertions in genome-wide motifs, normalized by subtracting the Tn5 bias from the footprinting signal. For each peak set, we employed CIS-BP motifs[98] (from *chromVAR* motifs human_pwms_v1) to calculate motif positions. Normalization of these footprints involved using mean values ± 200–250 from the motif center, after which we plotted the mean and standard deviation for each footprint pseudo-replicate. Comparison of the TF footprint between groups was conducted using the Wilcoxon rank sum test or one-way ANOVA followed by Tukey's multiple comparison test. A *p*-value less than 0.05 was considered statistically significant.

**scATAC-seq peak to gene linkage analysis**. We utilized the 'addPeak2GeneLinks' function in *ArchR* to predict peak-to-gene links, setting the 'corCutOff' parameter to 0.4 and 'reducedDims' to the dimensionality reduction. The resulting 'GRanges' object was utilized for visualization.

**Autoimmune SNPs analysis**. Pre-computed fine-mapped autoimmune-disease-associated SNPs were downloaded from Farh et al.[31]. We used the g-chromVAR[74] algorithm to identify enrichment of disease variants in each cell type. In brief, the summary statistics we downloaded were converted to hg38 coordinates using the UCSC liftover tool (v377) and formatted for *g-chromVAR*[74]. The methodology of *g-chromVAR* was previously described in detail[74]. Briefly, *g-chromVAR* weights chromatin accessibility features by fine-mapped GWAS variants posterior probabilities and calculates the enrichment for each cell type feature intensity. We first binarized the scATAC-seq dataset matrix with one column per cell type. We then followed the recommended guidelines for GWAS enrichment using 'computeWeightedDeviations' with default parameters. We then applied the Mann–Whitney *U* test and the Benjamini-Hochberg procedure for multiple-testing correction to compute enrichment *p* values[101].

**scRNA-seq processing**. All the samples were processed with the Chromium Single Cell Library, Gel Bead, and Multiplex Kit, and Chip Kit (10x Genomics) to barcode and convert the libraries on the 10x Genomics chromium platform. The Single-cell RNA libraries were prepared using the Chromium Single Cell 5 v2 Reagent (10x Genomics, 1000263) kit, following the manufacturer's instructions. For sequencing, the scRNA-seq libraries were sequenced on Illumina NovaSeq6000 in pair-end mode, and their quality was checked using *FastQC* software. Raw data was processed and aligned o the GRCh39 reference by the *cellranger* software with default parameters (https://support.10xgenomics.com, version 3.1.0) for each sample.

To demultiplex and barcode the sequences obtained from the 10x Genomics single-cell RNA-seq platform, we employed the 'cellranger-count' function in the *cellranger* Software Suite (10x Genomics). To aggregate all the samples, we used 'cellranger-count' function.

**scRNA-seq quality control**. For quality control, we filtered the low-quality cells with greater than 11% of mitochondrial genes and fewer than 200 or more than 3000 detected genes using *Seurat* V3. We further filtered the cell populations identified as red blood cells and platelets that expressed *HBB*, *HBA1*, *PPBP*, and *PF4* genes[17].

**scRNA-seq dimensionality reduction and clustering**. Downstream analysis of scRNA-seq dataset was performed using *Seurat* v3[69] as previously described[17]. To account for technical noise, we

choose the top 5000 most variable genes calculated by 'FindVariableFeatures' function were used for normalization and scaled. We performed principal component analysis (PCA) on the highly variable genes. The first 30 principal components were further analyzed. We then performed cell clustering based on KNN graphs using the 'FindNeighnors' and the 'FindClusters' with resolution set as 0.8 in *Seurat*. We did not use any batch correct method. We further performed the UMAP analysis, a dimensionality-reducing visualization tool, was used to embed the dataset into two dimensions. We did not detect potential batch effects in our dataset. Therefore, no batch correction method was applied in our further analysis.

**scRNA–seq differential analysis**. For scRNA-seq differential expression analysis, we used the "FindAllMarkers" function from the *Seurat* package with default parameters. Wilcox rank-sum test was used. The DEGs with logFC > = 0.25 were shown. A *p* value of less than 0.05 was considered statistically significant. To validate our results of differential analysis between BD and nonBDs, we also used the *Muscat* R package[36]. We followed the *Muscat* tutorial, and used the 'aggregateData' function to aggregate our scRNA count assay by cell clusters and samples. The R package *limma*[102] was to make a contrast matrix for BD and nonBDs. Once we have assembled the data, we used the 'pbDS' function with parameters set as default. In line with the *Seurat* method, DEGs with logFC > = 0.25 were shown. A *p* value of less than 0.05 was considered statistically significant. The R package *GeneOverlap*[103] was utilized to identify (1) the number of overlapping DEGs from *muscat* method and *Seurat* method, and (2) the statistical significance of this overlap based on list size and total number from both methods.

**scRNA–seq cellular annotation**. To validate our manual annotation, we first downloaded the PBMC multimodal single-cell dataset[23] as a reference and used the R package *clustifyR*[68] to perform cellular annotation. We chose the 'celltype. l2' from the reference as our label and calculated the average expression of the assay of the reference as input reference data. Then we followed the tutorial of *clustifyR* website with the parameter set as default.

**Multiomics data processing**. To integrate scRNA-seq and scATAC-seq dataset, we followed the pipeline outlined on ArchR[24], Seurat[69] and Signac[70] websites. First, ArchR[24] was used to split the complete dataset into smaller subsets of cells, enabling separate alignments and reducing computational RAM. We then used Seurat's canonical correlation analysis (CCA) to integrate the epigenetic and transcriptomic data. No additional batch correction methods were implemented. For this integration, the log-normalized and scaled scATAC-seq gene score matrix was aligned with the scRNA-seq gene expression matrix. By directly aligning cells from scATAC-seq with cells from scRNA-seq, the union of the 2,000 most variable genes in each modality as input for Seurat's "FindTransferAnchors" function and "TransferData" function, using default parameters. To find the nearest neighbor cell in the other modality for each profiled scRNA-seq and scATAC-seq cell, nearest-neighbor search was conducted in the joint CCA L2 space. These modality-spanning nearest-neighbor cell matches from all gestational timepoints were then combined to obtain dataset-wide cell matching.

**Gene-regulatory network workflow**. We used FigR to infer transcriptional regulators of target genes and construct GRN. We first used the single-cell peaks matrix from scATAC-seq and the count matrix from scRNA-seq (NscATAC-seq = 150,000 cells, NscRNA-seq = 150,000 cells, cells were sampled from the dataset)

and used the "runGenePeakcorr" with hg38 genome to determine cis-regulatory associations. Next, we filtered the correlations that $p > 0.05$ and defined DORC genes as those with more than seven significant peak-gene associations. We used the "runFigRGRN" function with scRNA-seq matrix and DORC data to generate GRN and selected TFs using ggplot2 to visualize the data.

**Statistics and reproducibility.** Statistical analysis of the frequencies of immune cell subpopulations between groups was performed using two-sided pairwise Wilcoxon test with Bonferroni's post-hoc correction with GraphPad Prism 8.0. Two-sided $p$ values of less than 0.05, were considered statistically significant. All the statistical details for the statistical tests can be found in the figure legends as well as in the Method Details section. In estimating the GO biological process and pathway, $p$ values were derived by a hypergeometric test with the default parameters in the Metascape webtool[104]. Each figure legend includes the details of the size of biological replicates and the assays. Values of $p <= 0.05$ were considered statistically significant. $p$ values are denoted as $*p <= 0.05$, $**p <= 0.01$, $***p <= 0.001$, $****p <= 0.0001$ in the figures. The statistical tests employed are referred to in the respective figure legends.

**Reporting summary.** Further information on research design is available in the Nature Portfolio Reporting Summary linked to this article.

## Data availability

The scRNA-seq, scATAC-seq and bulk RNA-seq data analyzed in the article are available from the corresponding author upon request under the Project Accession No. PRJCA004696 and the GSA Accession No. HRA004778 in https://ngdc.cncb.ac.cn/gsa-human/. Source data underlying Figs. 1c, 2c, 4j data are provided in Supplementary Data 2; Source data underlying Fig. 2e data is in Supplementary Data 4; Source data underlying Figs. 3c, g, h, 4e, Supplementary Figs. 8f, 9c, 10g, data are in Supplementary Data 7; Source data underlying Figs. 3d, 4f, k data are in Supplementary Data 6; Fig. 4i data is in Supplementary Data 8; Source data underlying Figs. 2d, 4d, l are in Supplementary Data 3; Source data underlying Supplementary Fig. 7a is in Supplementary Data 5; Source data underlying Fig. 6b is in Supplementary Data 10; Source data underlying Fig. 6e is in Supplementary Data 11.

## Code availability

All custom code used in this work is deposited in Zenodo https://doi.org/10.5281/zenodo.8348340[105].

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

## Acknowledgements

This study was funded by the National Natural Science Foundation of China (81721003); Local Innovative and Research Teams Project of Guangdong Pearl River Talents Programme (2017BT01S138); CAMS Innovation Fund for Medical Sciences (2019-I2M-5-005); the State Key Laboratory of Ophthalmology, Zhongshan Ophthalmic Center, Sun Yat-sen University. The funding body had no role in study design, collection, management, analysis and interpretation of data, writing of the manuscript, and the decision to submit the manuscript for publication.

## Author contributions

Y.Z., Y.L. and L.D. designed research; W.S. and J.Y. analyzed data; Z.S., C.P., Q.Z. and Y.L. processed the human samples and library construction; W.S. wrote the paper. X.L. provided comments and reviews during the revision process.

## Competing interests

The authors declare no competing interests.
