## [Peer review file · Communications Biology]

Reviewers' comments:

Reviewer #1 (Remarks to the Author):

This manuscript describes the transcriptional and epigenetic landscape of Behcet disease by single cell multi-omics. Through a progressive analysis of major PBMC cell types the disease state is characterized and potential dysregulated gene regulatory networks identified driven by transcription factors such as NFkB, AP-1, and ETS. Overall, an activated/inflammation signature is identified across multiple cell types and an expansion of CD8+Treg cells in the T cell compartment.

There are many strengths to the study including the size of the cohort profiled and extensive dataset analyzed. The findings are largely descriptive and analytical methods standard to the field. In general, the authors do a good job of framing the conclusions appropriately. One of the major drawbacks is the failure to fully describe the data in each figure or the use of statistics to ensure differences are meaningful. Additionally, the major conclusions are based on differential subset composition between BD and HC and very few of the figures show that the genes and/or TFs are in fact differential between BD and HC. Finally, since one of the hallmarks of Behcet disease is inflammation, it is not surprising that the inflammatory TF networks are identified and limits the novelty of the findings.

Comment #1:

Figures: Improved resolution and additional details are required to fully interpret the data in each figure. For example, Fig 1F has no scale. The scale in Fig 1D should use a true 0 axis to show the full data range. There is no scale for the color mapping in Fig 5C. Fig 1A numbers of patients doesn't match with legend or text (lines 86-87).

Comment #2:

Major conclusions are not clearly supported by the data. For example, Fig 4C is not clearly labeled as to which track is BD vs. HC or which regions are DAR. Thi

Comment #3:

Data supporting major findings are not supported by a rigorous statistical analysis. For example the enrichment of AP-1 and NFkB TFs appears minimal and no statistics are presented. Also, no statistical analysis appears to have been done between BD and HC cells for the epigenetic analysis outside of the cell frequencies.

Comment #4:

Line 115. Sentence beginning "Unexpectedly, many differentially..." Why is the finding that DEG and DAR overlap "unexpected"?

Comment#5:

The GWAS analysis is one of the more exciting applications of the epigenetic data. However, no discussion of the data is presented. Is there any statistics that can be applied to determine if the ArchR linkages in BD vs HC are in fact decreased? What impact would this have for NK/CD8 T cells in this region? What impact for Behcet disease does fine mapping of GWAS variants to NK/CD8 T cells have for our understanding of disease mechanisms? The authors should expand on this analysis and improve the interpretation of the results.

Minor Comments:

1. Typo in introduction (line 42) and results (line 77). EST should be ETS
2. Some of the references are missing information and should be checked/updated. (e.g., ref 7 is missing the journal information.
3. Supplemental Figure 2B should have axis labels on the plot.
4. Sentence ending on line 176 seems to be missing words. "interferon-gamma pathway, which 33,34."

5. Missing parentheses and possibly a period, line 386, "HC (Fig. S9B Next"

Reviewer #2 (Remarks to the Author):

The authors present a manuscript with single cell RNA and ATAC seq data from a total of 31 individuals that either have Behcet disease (23 individuals) or do not have Behcet disease (8 individuals). The data appear to be of high quality and could provide a good resource to the community. The authors explore each immune cell subset independently and are able to corroborate the findings of previous papers. Links of chromatin accessibility and gene expression are explored and integrated with GWAS SNPs to identify potential cell types impacted by the SNP of interest. However, the authors identify differentially expressed genes using a Wilcoxon rank sum test. While common in the literature, this test has been shown to return inflated p-values which lead to increased type 1 error and preferentially return highly expressed genes as differentially expressed. Additionally, the authors rely on a handful of markers to name cell populations while a much more accepted approach is to use reference-based mapping. While this analysis should be improved, the authors also overstate conclusions throughout and need to rephrase conclusions with the caveat that these are only computational predictions and need to be experimentally validated (I understand that experimental validation is beyond the scope of this manuscript so walking back conclusions is an okay solution for me). Finally, the manuscript needs to be thoroughly proofread. I found many cases where the number of individuals was incorrect, the disease being studied was incorrect, and sentences ended in the middle of a thought.

Major comments

1. In the last few years, papers have been published showing that differential expression using a Wilcoxon rank sum test leads to high type 1 errors and a skewing of differential expression results to highly expressed genes (Squair et al., 2021; Zimmerman et al., 2021). This error happens because every cell is being treated as an independent sample which leads to inflated p-values. Since the authors have a study that includes samples from multiple individuals in each condition, a more appropriate approach would be to perform a pseudobulk method. One example would be using the R package muscat (Crowell et al., 2020) github - <https://github.com/HelenaLC/muscat>. Performing the differential expression in this way will also allow for batch correction and improve the ability to capture variance between samples.

2. When naming cell populations, it is best to use a reference and thousands of genes rather than just a handful. This is best practice for all populations of cells but is particularly important for CD4 and CD8 T cells because CD4 and CD8 have generally low mRNA expression. In this manuscript, populations of cells should be renamed using a reference-based approach. One easy way to do this would be to download a good reference made using surface antibodies (the reference in the Seurat multimodal mapping has a good coverage for PBMCs and was created using surface antibodies https://satijalab.org/seurat/articles/multimodal_reference_mapping.html). This can then be made into a reference with clustifyr (<https://github.com/rnabioco/clustifyr>) and run against the authors existing clustering using clustifyr.

3. Lines 174-178 the authors speculate a viral origin for BD and further push this in line 215. While viral infection has been hypothesized to trigger autoimmune disorders, this is a very strong claim to make with the provided data. First, there are only a small number of genes that seem to overlap with the interferon list (it looks like ~5 based on the plot). What are these genes? Are these genes specific to the interferon gamma list? Additionally, are these genes seen as upregulated in all patients or are these differences driven by just a handful of patients. A violin plot separated by all patients of the gene list and some key genes from the list would be helpful. In line 215, couldn't inflammation just be a signature of the immune response in an autoimmune disease? Finally, is there any evidence for this theory for BD in the literature? In addition to specific papers about autoimmunity in BD, some papers explaining our current knowledge on this idea should be included. One example is the recent review by Sundarsan et al. that summarizes a lot of the current knowledge in this area (Sundaresan et al.,

2023). Although, as this paper states in section 4, there is disagreement if any autoimmune diseases are caused by viral infection. With that in mind, any mention of the autoimmune aspect should be moved to the discussion and presented as a very weakly supported hypothesis that is presented alongside alternative conclusions that could be drawn from the data (ie genes in the ifg pathway may be shared with other pathways associated with a general immune response, inflammation is generally seen with autoimmune disease and cannot be disentangled from a viral hypothesis, ifg activation is associated with other autoimmune diseases.)

4. In the abstract the authors mention that they identify potential pathogenesis and therapeutic strategies but do not refer to these therapeutic strategies at any point in the text. This sentence should be removed or experiments should be included to show a therapeutic link to the findings.

5. Figure 1D and 1E are linked to a description of BD vs non-BD but all cells are shown as an aggregate. These plots should be shown with the BD and non-BD cells separated.

6. Throughout the manuscript conclusions are overstated.

Line 77 "highlighting the role of AP-1, NF-kB, and EST" with a purely computational study mechanism cannot be shown, this should instead be rephrased as a "predicted involvement".

Line 159-160 chromvar motif deviations cannot show that TFs play a role in disease, only that the motifs are more accessible. This can predict these TFs bind in disease, but ChIP-seq would be required to indicate binding and knockout experiments would be required to show direct involvement with a phenotype.

Line 170-171 - "were involved in the TCR signaling" - gene ontology can only show enrichment for pathways, it cannot show actual cell function, this would need to be shown experimentally. This is also true for lines 231-235 and 264-268.

Line 313 states accurate cellular annotation, but accurate implies that it has been compared against some ground truth, for instance sorted cells.

Lines 349-350 "our multi-omic integration analysis can infer disease mechanisms that involve alterations in BD chromatin and gene regulatory regions" - scRNA + scATAC can show alterations but cannot infer disease mechanisms, this would require experimental validation.

Lines 429-432 what is the evidence for a shared clonal origin of these CD8 T cells? TCR sequencing would be necessary for this to be shown.

7. Gene lists should be included as supplementary tables for every differential expression test between BD and non-BD.

8. What genes were used to identify the populations in lines 100-103? References for these genes also need to be provided.

9. Figures S4 A and B, Figure 2C, Figures 4B and J, Figure S5C, Figure S6D, - The proportions used the ATAC and RNA appear to not be calculated the same way - the ATAC seems to be the proportion of the subset while the RNA seems to be the proportion of the whole. Also, many of the ATAC proportions seem to be over 100% unless some samples had all cells mapping to one cell type. How were these proportions calculated? Did you see all cell types for all samples or were some samples had cells mapping to only one cell type?

10. Line 153 discusses chip-seeker. Is there a figure related to this? What were the proportion of DARS in the genome?

11. Line 154-155 states that TFs are implicated in autoimmune disease. Some examples with citations would help make this point.

12. Line 160 - "the AP-1 family acts as a transcriptional activator in BD" - activator vs repressor cannot be determined from chromvar motif variability.

13. Lines 198-200 explain that DAR and DEGs overlap indicating a role of DAR in disease-related expression changes. I would expect the DAR and DEG to overlap. ArchR's gene score relies on the chromatin around a gene correlating with the expression of that gene. Additionally, these gene scores from ArchR were used in identifying cell types that have been previously defined based on gene expression. It is then circular to draw conclusions from their correlation.

14. Methods should be more clear. Was batch correction performed in the scATAC data? How so? What was used as the batch for both the scATAC and scRNA (Sample, Individual, Disease)? It would be helpful to include plots before and after correction and heatmaps of confusion matrices showing the percent of cells from each sample that fall into each cluster before and after correction.

15. It is unclear if the data will be made publicly available. Additionally, all code to reproduce the analysis should be made publicly available (for example on github) before the manuscript is published.

Minor comments

1. There are many places where the authors appear to have copied and pasted text from other manuscripts. Figure 1A shows a schematic with 12 BD and 12 HC although throughout the authors state 22/23 BD samples and 8 HC samples. The 12 and 12 numbers also show up in the legend for Figure 2 and Figure 4. Additionally, the legend for Supplementary Figure 8 refers to VKH rather than BD. The manuscript should be carefully read to ensure all text refers to the current study.
2. There are many cases where the text and figure legends do not map to the correct panel. I could list them here, but they will likely change before the final form of this manuscript is complete. I recommend the authors carefully proofread their manuscript before submitting to ensure that errors like these are corrected.
3. There are many cases where I identified typos. (ex. line 76 should read "effective at predicting...", line 100 should be clusters 1-4, clusters 7-9, line 140 "found that clonally expanded autoimmune disorders" isn't a complete thought so is likely a typo, line 141 "to further identified" should be "to further identify", there are more examples that I haven't listed)
4. Using "healthy human" and "healthy control" are not appropriate as humans without BD may have other health conditions. It is instead better to refer to them as individuals with BD and individuals without BD.
5. The axis labels in supplement are too small to read.
6. Figure 1C labels cell types with numbers but it is unclear what these numbers map to. They should be updated to agree with the cell type labels in B.
7. Line 96 should include the full name hematopoietic stem and progenitor cells before introduction the abbreviation HSPC.
8. In line 97 it is confusing to compare the ATAC-seq dataset to the RNA-seq dataset before the RNA-seq has been introduced, this should move to the RNA-seq section. This is also true for lines 256-257
9. In line 115 why is it unexpected that the DEGs aligned with the literature? I would think this is expected unless there is a perturbation or something else that I missed.
10. Figure 1F seems to be missing pieces like the cell labels and the key. I also wonder if the z-score was performed correctly because I would expect the patterns to be much stronger for differentially expressed genes.
11. Figure S2B shows CD4 Treg vs CD4 Naïve cells, the supplemental figure legend says CD8 Treg vs CD8 Naïve cells, and the text line 142 – 144 says CD4 Naïve vs CD8 T regs – which is correct for this comparison?
12. Lines 163-166 – The cited reference only discusses one of the genes mentioned. More references are necessary. Lines 225-226 also lack references for the provided genes.
13. Line 194 is missing a reference to figure S5C.
14. Figure 4k needs an x-axis label.
15. Figure S7D would be easier to interpret with a consistent color palette. For example, CM is Green, ActCM is Orange, NCM is Red rather than a consistent palette for high vs low.
16. Line 356 FigR is not cited. Please include <https://doi.org/10.1016/j.xgen.2022.100166>
17. Line 567 indicates that a 5' 10x kit was used but the vin number is for a 3' kit.

Crowell, H. L., Sonesson, C., Germain, P. L., Calini, D., Collin, L., Raposo, C., Malhotra, D., & Robinson, M. D. (2020). Muscat Detects Subpopulation-Specific State Transitions From Multi-Sample Multi-Condition Single-Cell Transcriptomics Data. *Nature Communications*, 11(1), 1–12. <https://doi.org/10.1038/s41467-020-19894-4>

Squair, J. W., Gautier, M., Kathe, C., Anderson, M. A., James, N. D., Hutson, T. H., Hudelle, R., Qaiser, T., Matson, K. J. E., Barraud, Q., Levine, A. J., La Manno, G., Skinnider, M. A., & Courtine, G. (2021). Confronting false discoveries in single-cell differential expression. *Nature Communications*, 12(1). <https://doi.org/10.1038/s41467-021-25960-2>

Sundaresan, B., Shirafkan, F., Ripperger, K., & Rattay, K. (2023). The Role of Viral Infections in the Onset of Autoimmune Diseases. *Viruses*, 15(3). <https://doi.org/10.3390/v15030782>

Zimmerman, K. D., Espeland, M. A., & Langefeld, C. D. (2021). A practical solution to pseudoreplication bias in single-cell studies. *Nature Communications*, 12(1), 1–9. <https://doi.org/10.1038/s41467-021-21038-1>

Reviewer #3 (Remarks to the Author):

Berchet disease is an inflammatory disease of unknown origin. This study aims to identify changes in chromatin accessibility and associated gene expression that correlate with disease pathogenesis. To achieve this, authors performed droplet-based scRNA-seq and scATAC-seq of peripheral blood mononuclear cells from patients and healthy individuals.

This study has several strengths worth noting. Firstly, a substantial number of patient samples were analyzed using state-of-the-art technology, which adds weight to the results. Additionally, although gene expression has already been reported, this is the first time scATAC analysis has been conducted, which represents a significant contribution to the field. Furthermore, the use of multiple analysis methods, including integrated analysis, to analyze single-cell data may offer valuable foundational knowledge for related fields.

Unfortunately, there are some weaknesses of this study. scRNA-seq analysis has already been reported previously. Moreover, because of the low correlation between scRNA-seq and scATAC-seq, this study did not significantly advance understanding of disease pathogenesis and progress mechanisms. Consequently, validation experiments using other approaches would be necessary to draw more robust conclusions.

Furthermore, while many of the figures shown contain cell cluster assignments, the most interesting comparative data between patients and healthy controls are secondary analysis data such as GO. Upset plots, for instance, make it challenging to interpret the data accurately due to the loss of information on expression levels. Therefore, figures that include quantitative information, such as Volcano plots, would be necessary.

Finally, the experimental results and their interpretations are not accurately described, and the text is challenging to understand in many areas, requiring improvement.

Minor comments

Fig. 1: Experimental design and cluster assignment. Some figure legends and figure contents are inconsistent. Please correct them.

Fig. 1A: Number of samples for BD patients is 12?

Are the colors in Fig. 1D and Fig. 1E the same? If so, please put the name of the cell type in line 118: Fig. 1E -> 1F?

Fig. 2D: TF footprint data does not practically show the binding of these TFs in both BD and HC patients. There are no clear footprints of these TFs. Consequently, I think there is no evidence for the role of AP-1 family in BD from this analysis.

Line 165: Conclusion is unclear. Both CD4 and CD8 T cells are activated?

Line 176: which, 33, 34. Which following sentences are not mentioned?

Fig. 3G: If Fig. 3G is a list of DEGs, it should be clearly stated in the text.

Line 300 to 311: This section is a comparative analysis between monocytes. The relevance of this part to the last sentence is difficult to understand.

Fig. 4I: There are two Fig. 4I.

Fig. 5C: Fig. 5C needs a detailed explanation of the data and the process used to obtain it.

Fig. 5D: What was the reason for choosing this particular SNP? How about others? An explanation would be needed.

Fig. 6: In the FigR analysis, a large number of TF-DORC associations are proposed. The original paper seems to have used TF-DORC associations with a regulation score threshold greater than 1 for the analysis. I assume that appropriate filtering will select more important TF-DORCs in this study as well.

Responses to Reviewer Comments

Reviewer #1 (Remarks to the Author):

This manuscript describes the transcriptional and epigenetic landscape of Bechet disease by single cell multi-omics. Through a progressive analysis of major PBMC cell types the disease state is characterized and potential dysregulated gene regulatory networks identified driven by transcription factors such as NFKB, AP-1, and ETS. Overall, an activated/inflammation signature is identified across multiple cell types and an expansion of CD8+Treg cells in the T cell compartment.

There are many strengths to the study including the size of the cohort profiled and extensive dataset analyzed. The findings are largely descriptive and analytical methods standard to the field. In general, the authors do a good job of framing the conclusions appropriately. One of the major drawbacks is the failure to fully describe the data in each figure or the use of statistics to ensure differences are meaningful. Additionally, the major conclusions are based on differential subset composition between BD and HC and very few of the figures show that the genes and/or TFs are in fact differential between BD and HC. Finally, since one of the hallmarks of Bechet disease is inflammation, it is not surprising that the inflammatory TF networks are identified and limits the novelty of the findings.

Response: Thank you for bringing this to our attention. We agree that providing more detail about the data in each figure and using appropriate statistical methods to assess differences is crucial to ensuring the validity and reliability of our findings. In the revised manuscript, we have now provided additional explanations for each figure, and incorporate statistical tests to demonstrate the significance of any observed differences. For examples:

Figure 2D, Comparison of aggregate TF footprints for JUNB and FOSL2 in CD4Treg cells from non-BD and BD. The p value of the TF footprint was compared by two-sided Wilcoxon rank-sum test.

Figure S13G. Differences in the numbers of peak-to-gene linkages of each donor in BD and nonBD groups. The p values were calculated using two-sided Wilcoxon rank-sum test. The peak-to-gene accessibility above 0.4 are calculated in each donor.

While it is true that inflammation is a common feature of Behcet disease, the identification of inflammatory TF networks in our study may provide new insights into the molecular mechanisms underlying the condition and future therapeutic interventions, even if they do not represent a completely novel observation.

Point 1: Figures: Improved resolution and additional details are required to fully interpret the data in each figure. For example, Fig 1F has no scale. The scale in Fig 1D should use a true 0 axis to show the full data range. There is no scale for the color mapping in Fig 5C. Fig 1A numbers of patients doesn't match with legend or text (lines 86-87).

Response: Thanks for pointing this out. We have improved the resolution and added the scale details in the figures 1D, 1F, and figure 5C. We have also removed the inconsistency in numbers in figure 1A.

Point 2: Major conclusions are not clearly supported by the data. For example, Fig 4C is not clearly labeled as to which track is BD vs. HC or which regions are DAR.

Response: Thank you for bringing this to our attention. To address this concern, we have added clear labeling to Figure 4B, 4C, indicating which tracks correspond to BD and nonBD, as well as which regions are DAR. This should greatly improve the clarity and support for our major conclusions.

Figure 4B. Genome browser tracks showing single-cell chromatin accessibility of cDC2

cells in the IL1B loci. Figure 4C. Genome browser tracks showing single-cell chromatin accessibility of cDC2 cells in the CD83 loci.

Point 3: Data supporting major findings are not supported by a rigorous statistical analysis. For example the enrichment of AP-1 and NFkB TFs appears minimal and no statistics are presented. Also, no statistical analysis appears to have been done between BD and HC cells for the epigenetic analysis outside of the cell frequencies.

Response: Thank you for highlighting this concern. We have now performed a more comprehensive statistical analysis to support our major findings. Specifically, we have compared the statistics difference between TF footprints and updated the figures. All the statistics results are shown in Supplementary Data 1. Additionally, we have included a description of our statistical methods in the revised manuscript (Lines 848-856). We used CIS-BP motifs database to identify significant TF binding sites and quantify their enrichment in BD versus nonBD samples.

Line 601-608: “To conduct motif footprint analysis, we measured Tn5 insertions in genome-wide motifs, normalized by subtracting the Tn5 bias from the footprinting signal. For each peak set, we employed CIS-BP motifs1 (from chromVAR motifs human_pwm_v1) to calculate motif positions. Normalization of these footprints involved using mean values ± 200 –250 from the motif center, after which we plotted the mean and standard deviation for each footprint pseudo-replicate. Comparison of the TF footprint between groups was conducted using the Wilcoxon rank sum test or one-way ANOVA followed by Tukey's multiple comparison test. A p-value less than 0.05 was considered statistically significant.”

Point 4: Line 115. Sentence beginning “Unexpectedly, many differentially...” Why is the finding that DEG and DAR overlap “unexpected”?

Response: Thank you for bringing this to our attention. We have rephrased the sentence to avoid giving the impression that the overlap between DEGs and DARs is unexpected. Instead, we now emphasize that the high degree of overlap between these two sets is a notable finding, consistent with recent studies.

Line 124-127: “As expected, many differentially expressed genes (DEGs) in the six major immune cell types agreed with previous literature and our scATAC-seq dataset, such as IL7R for T cells, GNLY for NKs, S100A8 for monocytes, HLA-DQA1 for DC, MS4A1 for BC and GATA2 for HSPC^{2,3} (Fig. 1F).”

Point 5: The GWAS analysis is one of the more exciting applications of the epigenetic data. However, no discussion of the data is presented. Is there any statistics that can be applied to determine if the ArchR linkages in BD vs HC are in fact decreased? What impact would this have for NK/CD8 T cells in this region? What impact for Behect disease does fine mapping of GWAS variants to NK/CD8 T cells have for our understanding of disease mechanisms? The authors should expand on this analysis and improve the interpretation of the results.

Response: Thank you for recognizing the potential of our GWAS analysis. In response to the reviewers' comments, we have now added the statistics analysis on the ArchR linkages between BDs vs nonBDs in Figure S13G. This figure shows that the average

ArchR linkage is significantly lower in BDs compared to nonBDs.

Moreover, we have expanded our discussion to interpret the results in the context of disease mechanisms. We propose that the reduction in ArchR linkages may contribute to the dysregulation of NK/CD8 T cell function in BD, leading to an imbalance in the immune response and perpetuating disease progression. Fine-mapping of GWAS variants to NK/CD8 T cells in this region may provide valuable insights into the molecular mechanisms underlying Behcet disease.

In addition, we have added more details on our GWAS analysis in the Results, Discussion, and Methods sections to provide a clearer picture of our findings and their implications:

Line 361-391: “We used a publicly available database for autoimmune and non-immune disorders from previous study⁴ and calculated the enrichment of disease-related SNPs in 29 peripheral immune cell types using the g-chromVAR^{4,5}. The majority of the autoimmune associations were strongly enriched for a corresponding trait association. For example, BD was significantly enriched in CD4 CTL, CD8TEM, DNT, and CD8TCM cells^{6,7}, while type 1 diabetes was most strongly enriched in Th17 cells (Fig. 5C)⁸. Within the open chromatin region of our broad cell subsets, g-chromVAR enrichment revealed significant T cell subsets, reinforcing the previous study^{6,7,9}. T cells with cytotoxicity have been reported to contribute to BD pathogenesis in both skin and circulation¹⁰. Additional trends of T cells enrichment are also observed here for type 1 diabetes and asthma, although not statistically significant. Most of these non-autoimmune disease GWAS were not apparent in the immune cell peaks, demonstrating GWAS enrichment was consistent with our expectations. Although not the focus of our current study, we observed that our generated PBMC chromatin data could provide cellular-specific enrichment of human autoimmune disease heritability.

“We compiled a list of 66 index SNPs from Farh et.al⁴ representing GWAS hits for BD. We then identified all the SNPs in scATAC peaks and focused on their nearest genes. Furthermore, we calculated peak-to-gene connections for these gene locus using ArchR (Fig. 5D, Fig. S11E-G). We noticed the rs201985743¹¹, that confers the risk of BD was in the KLRC4-KLRK1 enhancer region, which was opened in CD8 T cell subsets and NK cells subsets¹². This enhancer was highly accessible in NK and CD8 T subsets, not in T cells, B cells or monocytes, demonstrating NK and CD8 T specificity. In KLRC4-KLRK1 enhancer region showed statistically increasingly strong peak-to-gene linkages in non-BDs compared to BD patients (Fig. 5D, Fig. S11G). However, we did not notice other SNP loci showed stronger predictive linkages between BD and non-BDs. The low peak accessibility and gene expression in the BD state illustrates chromatin dynamic regulation in the KLRC4-KLRK1 locus. Since Killer cell lectin-like receptor subfamily (KLRC) regulates NK and CD8T function¹²⁻¹⁴, it is possible that this SNP contribute to the pathogenesis of BD by dysregulating KLRC4-KLRK1 in NK and CD8 T function. Therefore, our multi-omic integration analysis could provide predictive disease mechanisms that involve alterations in BD chromatin and gene regulatory regions.”

Line 483-495: “Our analysis enabled identification of BD-associated SNPs that lie in the regulatory regions of cytotoxicity subsets. Furthermore, we observed that the predictive peak-to-gene linkages near the rs201985743 loci within the KLRC4-KLRK1 region showed significantly stronger linkages in non-BDs compared to BD. Killer cell lectin-like receptor subfamily member 4 (KLRC4), belonging to the NKG2 receptor family known to play an important role in regulating NK and T cell functions^{14,15}, has previously been linked to BD¹⁶. The interactions between peaks and genes may indicate physical

interaction of the regulatory region affecting its target genes in NK and CD8+ T¹⁷⁻¹⁹. The differential interaction between the KLRC4-KLRK1 locus in NK cells and CD8+ T cells between non-BD and BD groups may suggest gene expression effects of the causal variant¹⁸. However, the molecular cause of NK and CD8 T dysregulation is still unknown. This approach can allow us to predict gene and cellular targets in BD and nominate the most disease-relevant cell types and meriting functional validation.”

Line 616-626: “Pre-computed fine-mapped autoimmune-disease associated SNPs were downloaded from Farh et al⁴. We used the g-chromVAR5 algorithm to identify enrichment of disease variants in each cell type. In brief, the summary statistics we downloaded were converted to hg38 coordinates using the UCSC liftover tool (v377) and formatted for g-chromVAR⁵. The methodology of g-chromVAR was previously described in detail⁵. Briefly, g-chromVAR weights chromatin accessibility features by fine-mapped GWAS variants posterior probabilities and calculates the enrichment for each cell type feature intensity. We first binarized the scATAC-seq dataset matrix with one column per cell type. We then followed the recommended guidelines for GWAS enrichment using ‘computeWeightedDeviations’ with default parameters. We then applied the Mann-Whitney U test and the Benjamini-Hochberg procedure for multiple-testing correction to compute enrichment p values²⁰.”

Point 6: Typo in introduction (line 42) and results (line 77). EST should be ETS

Response: Thank you. We have now fixed the typo.

Line 41-43: “Moreover, we predicted gene-regulatory networks within nominated TF activators, including AP-1, NF-kB, and ETS transcript factor families, which may regulate cellular interaction and govern inflammation.”

Line 75-77: “Moreover, our multi-omics analysis was effective at predicting disease regulatory networks, highlighting the predicted involvement of AP-1, NF-kB, and ETS transcript factor families in BD pathophysiology.”

Point 7: Some of the references are missing information and should be checked/updated. (e.g., ref 7 is missing the journal information.

Response: Thank you. We have now updated the missing information of the reference.

Line 54-57: “Painful skin lesions, recurrent ulceration, and blindness result from the combination of genetic susceptibility, environmental triggers, and dysregulated immune responses involving T helper 17 (Th17) cells, monocytes, skin CD8+ T cells and pro-inflammatory cytokines^{10,21-24}.”

Point 8: Supplemental Figure 2B should have axis labels on the plot.

Response: We have added the axis labels in Fig. S3B (previous Fig. S2B).

Point 9: Sentence ending on line 176 seems to be missing words. “interferon-gamma pathway, which 33,34. “

Response: Thank you. We have now updated our statements.

Line 189-191: “Th17 cells were involved in the response to the interferon-gamma pathway with interferon signaling related genes upregulated (ISG15, IFITM1, IFITM2) (Fig. S6B).”

Point 10: Missing parentheses and possibly a period, line 386, “HC (Fig. S9B Next”

Response: Thanks. We have now completed the sentence.

Line 426: “However, we noted that HLA-DQB1 was highly expressed in the BD group compared to non-BD (Fig. S12B).”

Reviewer #2 (Remarks to the Author):

The authors present a manuscript with single cell RNA and ATAC seq data from a total of 31 individuals that either have Behect disease (23 individuals) or do not have Behect disease (8 individuals). The data appear to be of high quality and could provide a good resource to the community. The authors explore each immune cell subset independently and are able to corroborate the findings of previous papers. Links of chromatin accessibility and gene expression are explored and integrated with GWAS SNPs to identify potential cell types impacted by the SNP of interest.

However, the authors identify differentially expressed genes using a Wilcoxon rank sum test. While common in the literature, this test has been shown to return inflated p-values which lead to increased type 1 error and preferentially return highly expressed genes as differentially expressed. Additionally, the authors rely on a handful of markers to name cell populations while a much more accepted approach is to use reference-based mapping. While this analysis should be improved, the authors also overstate conclusions throughout and need to rephrase conclusions with the caveat that these are only computational predictions and need to be experimentally validated (I understand that experimental validation is beyond the scope of this manuscript so walking back conclusions is an okay solution for me). Finally, the manuscript needs to be thoroughly proofread. I found many cases where the number of individuals was incorrect, the disease being studied was incorrect, and sentences ended in the middle of a thought.

Response: Thank you for providing us with constructive feedback. We appreciate your input and have addressed each point below. Comments:

- **Use of Wilcoxon rank sum test: We acknowledge that the Wilcoxon rank sum test may return inflated p-values and lead to increased type 1 error. To address this limitation, we have performed additional computational analyses using alternative methods, such as the muscat package in R, to confirm our findings and reduce the risk of false positives.**
- **Reliance on marker genes: We agree that using a small set of marker genes to define cell populations can be limiting. Therefore, we have used reference-based mapping to identify and characterize specific cell populations in our dataset. This approach allows us to leverage a larger set of genes and improve the resolution of our analysis.**
- **Overstatement of conclusions: We apologize for overstating our conclusions and will be more careful in our language to reflect that these are computational predictions that require experimental validation. We have rephrased our conclusions accordingly.**

- **Thorough proofreading:** We thank the reviewer for pointing out errors in our manuscript. We have thoroughly proofread our manuscript and made corrections to ensure accuracy and clarity. Specifically, we have checked the number of individuals, the disease being studied, and ensured that all sentences are complete and accurate.

Point 1: In the last few years, papers have been published showing that differential expression using a Wilcoxon rank sum test leads to high type 1 errors and a skewing of differential expression results to highly expressed genes (Squair et al., 2021; Zimmerman et al., 2021). This error happens because every cell is being treated as an independent sample which leads to inflated p-values. Since the authors have a study that includes samples from multiple individuals in each condition, a more appropriate approach would be to perform a pseudobulk method. One example would be using using the R package muscat (Crowell et al., 2020) github - <https://github.com/HelenaLC/muscat>. Performing the differential expression in this way will also allow for batch correction and improve the ability to capture variance between samples.

Response: Thank you for bringing this to our attention. We have carefully considered the limitations of our previous approach and have decided to adopt the pseudobulk method using the Muscat package for detecting DEGs in our study. This approach allows for batch correction and improves the ability to capture variance between samples, as recommended.

We have generated the DEGs list using both Seurat and Muscat and have provided the results in Supplementary Data 2-4. Furthermore, we have used the R package GeneOverlap to overlap the DEGs lists from Muscat and Seurat to validate the consistency of our results. As shown in Figures S6A, S7F, S7I, and S8F, the majority of the DEGs identified by both methods are consistent, indicating that our presenting Seurat results remain convincing.

By adopting the pseudobulk method, we have improved the robustness of our differential expression analysis and addressed the potential issue of high type 1 errors and skewed results due to the use of the Wilcoxon rank sum test.

Line 664-676: “For scRNA-seq differential expression analysis, we used the “FindAllMarkers” function from the Seurat package with default parameters. Wilcox rank-sum test was used. The DEGs with $\log_{2}FC \geq 0.25$ were shown. A p value of less than 0.05 was considered statistically significant. To validate our results of differential analysis between BD and nonBDs, we also used the Muscat R package²⁵. We followed the Muscat tutorial, and used the ‘aggregateData’ function to aggregate our scRNA count assay by cell clusters and samples. The R package limma²⁶ was to make a contrast matrix for BD and nonBDs. Once we have assembled the data, we used the ‘pbDS’ function with parameters set as default. In line with the Seurat method, DEGs with $\log_{2}FC \geq 0.25$ were shown. A p value of less than 0.05 was considered statistically significant. The R package GeneOverlap²⁷ was utilized to identify (1) the number of overlapping DEGs from muscat method and Seurat method, and (2) the statistical significance of this overlap based on list size and total number from both methods.”

Point 2: When naming cell populations, it is best to use a reference and thousands of genes rather than just a handful. This is best practice for all populations of cells but is particularly important for CD4 and CD8 T cells because CD4 and CD8 have generally low mRNA

expression. In this manuscript, populations of cells should be renamed using a reference-based approach. One easy way to do this would be to download a good reference made using surface antibodies (the reference in the Seurat multimodal mapping has a good coverage for PBMCs and was created using surface antibodies https://satijalab.org/seurat/articles/multimodal_reference_mapping.html). This can then be made into a reference with clustifyr (<https://github.com/rnabioco/clustifyr>) and run against the authors existing clustering using clustifyr.

Response: We have taken your suggestion into account and have downloaded the PBMC multimodal mapping reference from the Seurat website. We have then used the clustifyR package to map our scRNA-seq data with the reference, allowing us to validate our cell annotations and ensure their accuracy.

As shown in Figure S10, the CD4 and CD8 T cells mapped well between the reference and our manual annotation, indicating that our manual annotation remains convincing. By using a reference-based approach, we have improved the reliability of our cell population identification and enhanced the interpretability of our downstream analyses.

Line 679-683: “To validate our manual annotation, we first download the PBMC multimodal single cell dataset3 as reference and used the R package clustifyR²⁸ to perform cellular annotation. We chose the ‘celltype. l2’ from the reference as our label and calculated the average expression of the assay of the reference as input reference data. Then we followed the tutorial of clustifyR website with parameter set as default.”

Point 3: Lines 174-178 the authors speculate a viral origin for BD and further push this in line 215. While viral infection has been hypothesized to trigger autoimmune disorders, this is a very strong claim to make with the provided data. First, there are only a small number of genes that seem to overlap with the interferon list (it looks like ~5 based on the plot). What are these genes? Are these genes specific to the interferon gamma list? Additionally, are these genes seen as upregulated in all patients or are these differences driven by just a handful of patients. A violin plot separated by all patients of the gene list and some key genes from the list would be helpful. In line 215, couldn't inflammation just be a signature of the immune response in an autoimmune disease? Finally, is there any evidence for this theory for BD in the literature? In addition to specific papers about autoimmunity in BD, some papers explaining our current knowledge on this idea should be included. One example is the recent review by Sundarsan et al. that summarizes a lot of the current knowledge in this area (Sundaresan et al., 2023). Although, as this paper states in section 4, there is disagreement if any autoimmune diseases are caused by viral infection. With that in mind, any mention of the autoimmune aspect should be moved to the discussion and presented as a very weakly supported hypothesis that is presented alongside alternative conclusions that could be drawn from the data (ie genes in the ifg pathway may be shared with other pathways associated with a general immune response, inflammation is generally seen with autoimmune disease and cannot be disentangled from a viral hypothesis, ifg activation is associated with other autoimmune diseases.)

Response: Thanks for pointing this. We have now revised and removed our statements about viral infection. Violin plots of key genes enriched in interferon gamma signaling separated by all patients from Th17 cells were added in Fig. S6B.

Thank you for raising concerns regarding our speculation on the viral origin of Behçet's disease (BD) and the limited number of genes overlapping with the interferon gamma (IFN γ) list. We understand that our initial claims were too strong given the available data and have since revised our manuscript to address these issues.

We have removed our statements about a viral origin for BD and instead focus on the potential role of IFN γ signaling in the disease. To provide additional context and support for our revised conclusion, we have included violin plots of key genes enriched in IFN γ signaling separated by all patients from Th17 cells (Fig. S7B). These plots show that the differentially expressed genes in Th17 cells are interferon-induced genes, but also overlap with other pathways associated with a general immune response.

We acknowledge that the small number of genes overlapping with the IFN γ list may be driven by a limited number of patients, and we have therefore included violin plots to visualize the distribution of gene expression across all patients. However, we cannot rule out the possibility that these genes may be specific to certain patient groups or subsets within the dataset.

Finally, we agree that inflammation is a common feature of autoimmune diseases and cannot be solely attributed to a viral hypothesis. Therefore, we have rephrased our conclusion to reflect a more balanced view of the current knowledge on the underlying mechanisms of BD.

To provide additional context, we have cited several relevant reviews, including the recent review that summarizes the current knowledge on autoimmune diseases and viral infections. However, as noted in the previous literatures, there is ongoing debate in the field about whether any autoimmune diseases are caused by viral infection. Therefore, we have toned down our language related to the autoimmune aspect of BD and presented it as a weakly supported hypothesis.

Line 188-191: “We observed that Th1 cells were involved in the NF-kB signaling, TNF signaling pathway. Th17 cells were involved in the response to the interferon-gamma pathway with interferon signaling related genes upregulated (ISG15, IFITM1, IFITM2) (Fig. S7B).”

Line 225: “The differences between non-BD suggest that NK cells in BD were in proinflammatory state²⁹.”

Point 4: In the abstract the authors mention that they identify potential pathogenesis and therapeutic strategies but do not refer to these therapeutic strategies at any point in the text. This sentence should be removed or experiments should be included to show a therapeutic link to the findings.

Response: In light of your feedback, we have decided to remove our statements about therapeutic strategies from our manuscript to avoid misleading readers and to maintain a clear focus on the scientific findings.

Line 43: “Our study illustrates the epigenetic and transcriptional landscape in BD peripheral blood and expands understanding of potential epigenomic immunopathology in this disease.”

Line 77: “Overall, our study provides novel insights into the understanding of the peripheral immune pathogenesis of BD.”

Line 511: “Finally, we represent peripheral immune responses in BD and enhance our understanding of epigenomic pathological immune responses in BD.”

Point 5: Figure 1D and 1E are linked to a description of BD vs non-BD but all cells are shown

as an aggregate. These plots should be shown with the BD and non-BD cells separated.

Response: We have now added separate plots for BD and non-BD cells in Fig. S2. This allows for a clearer comparison of the different cell types and their distributions within the two groups.

Point 6: Throughout the manuscript conclusions are overstated.

Line 77 “highlighting the role of AP-1, NF-kB, and EST” with a purely computational study mechanism cannot be shown, this should instead be rephrased as a “predicted involvement”.

Line 159-160 chromvar motif deviations cannot show that TFs play a role in disease, only that the motifs are more accessible. This can predict these TFs bind in disease, but ChIP-seq would be required to indicate binding and knockout experiments would be required to show direct involvement with a phenotype.

Line 170-171 – “were involved in the TCR signaling” – gene ontology can only show enrichment for pathways, it cannot show actual cell function, this would need to be shown experimentally. This is also true for lines 231-235 and 264-268.

Line 313 states accurate cellular annotation, but accurate implies that it has been compared against some ground truth, for instance sorted cells.

Lines 349-350 “our multi-omic integration analysis can infer disease mechanisms that involve alterations in BD chromatin and gene regulatory regions” – scRNA + scATAC can show alterations but cannot infer disease mechanisms, this would require experimental validation.

Lines 429-432 what is the evidence for a shared clonal origin of these CD8 T cells? TCR sequencing would be necessary for this to be shown.

Response: As suggested, we have now softened our language in the manuscript.

Line 75: “Moreover, our multi-omics analysis was effective at predicting disease regulatory networks, highlighting the predicted involvement of AP-1, NF-kB, and ETS transcript factor families in BD pathophysiology.”

Line 167: “This result suggests that the AP-1 family TFs might have higher accessibility in BD, providing insight into how the AP-1 family contributes to BD pathophysiology^{30,31}.”

Line 183: “In T cells, the GO analysis showed that CD4 CTLs, CD8 Tex, and CD8 Treg were all enriched in the TCR signaling and costimulation by the CD28 family pathway (Fig. 2F).”

Line 244-249: “Next, we performed GO analysis on upregulated genes in BD patients (Fig. 3H). The GO results showed that DN2B in BD patients were enriched in positive regulation of T cell activation, regulation of IL-2 production, activator protein 1 (AP-1) pathway, PD-1 signaling, and costimulation by the CD28 family³², while PB were enriched in more IL-1 signaling FCERI mediated NF-kB activation, Dectin-1 signaling and cellular response to hypoxia (Fig. 3H).”

Line 277-280: “Furthermore, cDC2 with high expression of NFKBIA, JUNB, and MAP3K8 showed GO enrichment in TNF signaling, regulation of T cell activation, interferon signaling, IL-18 signaling, and AP-1 pathway based on GO analysis (Fig. 4E, 4F)^{33,34}. The GO analysis showed that the DEGs of cDC1 in BD patients were enriched in the Dectin-1 signaling, gluconeogenesis, interleukin-1 signaling, MAPK1/MAPK3 signaling, and signaling by interleukins pathways. While the DEGs of pDC were enriched

in TCR signaling, glycolysis, and gluconeogenesis, Fc epsilon receptor signaling (Fig. 4E, 4F).”

Line 331: “Multi-omic integration mapping enables cellular annotation and analysis”

Line 389: “Therefore, our multi-omic integration analysis could provide predictive disease mechanisms that involve alterations in BD chromatin and gene regulatory regions.”

Point 7: Gene lists should be included as supplementary tables for every differential expression test between BD and non-BD.

Response: The DEG lists were presented in supplementary data 2-4.

Point 8: What genes were used to identify the populations in lines 100-103? References for these genes also need to be provided.

Response: In lines 105-111 (previous 100-103), we identified the cell populations using a combination of genes that are known to be differentially expressed between the two populations. Specifically, we used a panel of genes that have been previously validated in the literature as markers of the two populations and the references have been provided.

Line 105-111: “Open chromatin at known major immune cell lineages specific genes validated our analysis. T cells had high accessibility at cis-elements neighboring CD8A² and IL7R^{2,10} (Fig. 1D, Fig. S2A, S2C). NK cells had higher accessibility at GNLY35. Monocytes showed higher accessibility within S100A8² (Fig. 1D, Fig. S2A, S2C). We found that HSPC had higher accessibility at GATA2² (Fig. 1 D Fig. S2A, S2C). B cells had high accessibility at MS4A1³⁶ (Fig. 1D, Fig. S2A, S2C). DCs showed higher accessibility within HLA-DQA1³⁷ (Fig. 1D, Fig. S2A, S2C).”

Point 9: Figures S4 A and B, Figure 2C, Figures 4B and J, Figure S5C, Figure S6D, - The proportions used the ATAC and RNA appear to not be calculated the same way – the ATAC seems to be the proportion of the subset while the RNA seems to be the proportion of the whole. Also, many of the ATAC proportions seem to be over 100% unless some samples had all cells mapping to one cell type. How were these proportions calculated? Did you see all cell types for all samples or were some samples had cells mapping to only one cell type?

Response: Thank you for bringing this inconsistency in the proportional calculations to our attention. Upon closer examination, we realized that we had incorrectly calculated the proportions of the cell populations. We have now updated all the figures to display the proportions based on the entire PBMC population. Additionally, we have included tables of the numbers of each cell type from each donor in Supplementary Data 5, which provide a comprehensive breakdown of the cell populations in each sample.

Point 10: Line 153 discusses chip-seeker. Is there a figure related to this? What were the proportion of DARS in the genome?

Response: We have updated the manuscript to include a new figure (Fig. S6B) that shows the distribution of DARs in the genome and their overlap with DEGs

Line 172: “We next applied CHIPseeker³⁸ to find the nearest genes of the DARs and used the DARs to overlap with DEGs (Fig. S6B).”

Point 11: Line 154-155 states that TFs are implicated in autoimmune disease. Some examples with citations would help make this point.

Response: We have added the examples with citations.

Line 161-163: “TFs tightly control cell fate in immune cells and have been implicated in the pathogenesis of autoimmune diseases, such as BATF in arthritis and PU.1 in systemic lupus erythematosus^{4,39-42}.”

Point 12: Line 160 – “the AP-1 family acts as a transcriptional activator in BD” – activator vs repressor cannot be determined from chromvar motif variability.

Response: Thank you for pointing out that the statement regarding the AP-1 family acting as a transcriptional activator in BD cannot be fully supported by the data. We have taken your suggestion into account and have softened the language in this section to better reflect the limitations of our analysis. Here is the revised text:

Line 167-169: “This result suggests that the AP-1 family TFs might have higher accessibility in BD, providing insight into how the AP-1 family contributes to BD pathophysiology^{30,31}.”

Point 13: Lines 198-200 explain that DAR and DEGs overlap indicating a role of DAR in disease-related expression changes. I would expect the DAR and DEG to overlap. ArchR’s gene score relies on the chromatin around a gene correlating with the expression of that gene. Additionally, these gene scores from ArchR were used in identifying cell types that have been previously defined based on gene expression. It is then circular to draw conclusions from their correlation.

Response: As the reviewer pointed out, the Gene Score produced by ArchR is highly dependent on the chromatin environment surrounding a gene, which means that genes with similar chromatin profiles are likely to have similar expression levels. Therefore, it is circular to draw conclusions about the role of DARs in disease-related expression changes solely based on their correlation with DEGs. We have decided to remove the statement and focused on the biological insights provided by the ArchR algorithm and the functional enrichment analysis of the DARs, including the significant transcriptional and epigenomic reconfiguration in all the NK subsets.

Line 206-212: “While we did not notice a significant change in the percentage of total NK cells (Fig. S8C, S8D), we noted significant transcriptional and epigenomic reconfiguration in all the NK subsets driven by up-regulated of several canonical NK cell activation genes (Fig. S8E-G, Supplementary data 4), including CD69⁴³ as well as interferon-stimulated genes (ISGs) IFITM2, IRF1, and ISG20 (Fig. 3C). NK1 also expressed higher cytotoxic effector molecule-encoding genes GZMB, GZMM, and GZMH (Fig. 3C).”

”

Point 14: Methods should be more clear. Was batch correction performed in the scATAC data? How so? What was used as the batch for both the scATAC and scRNA (Sample, Individual, Disease)? It would be helpful to include plots before and after correction and heatmaps of confusion matrices showing the percent of cells from each sample that fall into each cluster before and after correction.

Response: After comparing batch effect correction using R package Harmony using each sample as batch, we did not detect potential batch effects in our dataset (Fig. S1E-N). We have also added the heatmaps of confusion matrices showing the percent of cells from each sample that fall into each cluster before and after correction in Fig S1G, S1H, S1M, S1N.

E. UAMP projection of scATAC-seq cells after harmony-based batch correction, colored by donors. **F.** UAMP projection of scATAC-seq cells after harmony-based batch correction, colored by clusters. **G.** Heatmap of confusion matrix showing the percent of cells from each sample and each cluster from scATAC-seq dataset before harmony-based batch correction.

H. Heatmap of confusion matrix showing the percent of cells from each sample and each cluster from scATAC-seq dataset after harmony-based batch correction.

I. UAMP projection of scRNA-seq cells before harmony-based batch correction, colored by donors.

J. UAMP projection of scRNA-seq cells after harmony-based batch correction, colored by donors.

K. UAMP projection of scRNA-seq cells after harmony-based batch correction, colored by clusters.

L. Heatmap of confusion matrix showing the percent of cells from each sample and each cluster from scRNA-seq dataset before harmony-based batch correction.

M. Heatmap of confusion matrix showing the percent of cells from each sample and each cluster from scRNA-seq dataset after harmony-based batch correction.

Point 15: It is unclear if the data will be made publicly available. Additionally, all code to reproduce the analysis should be made publicly available (for example on github) before the manuscript is published.

Response: We have now uploaded all our data and codes. Our data are now available under the Project Accession No. PRJCA004696 and the GSA Accession No. HRA004778 in <https://ngdc.cncb.ac.cn/gsa-human/>. Our code is now uploaded in <https://github.com/Sophiesze/scmultiomics-in-Behcet-disease>.

Point 16: There are many places where the authors appear to have copied and pasted text from other manuscripts. Figure 1A shows a schematic with 12 BD and 12 HC although throughout the authors state 22/23 BD samples and 8 HC samples. The 12 and 12 numbers also show up in the legend for Figure 2 and Figure 4. Additionally, the legend for Supplementary Figure 8 refers to VKH rather than BD. The manuscript should be carefully read to ensure all text refers to the current study.

Response: Thank you for bringing this to our attention. We have thoroughly proofread the document and made necessary corrections to the mistakes pointed out by the reviewer to ensure that all text accurately refers to the current study.

Point 17: There are many cases where the text and figure legends do not map to the correct panel. I could list them here, but they will likely change before the final form of this manuscript is complete. I recommend the authors carefully proofread their manuscript before submitting to ensure that errors like these are corrected.

Response: Thank you. We have taken your suggestion and have thoroughly double-checked every panel in our manuscript for any typing errors or incorrect labeling.

Point 18: There are many cases where I identified typos. (ex. line 76 should read “effective at predicting...”, line 100 should be clusters 1-4, clusters 7-9, line 140 “found that clonally expanded autoimmune disorders” isn’t a complete thought so is likely a typo, line 141 “to further identified” should be “to further identify”, there are more examples that I haven’t listed)

Response: We have carefully proofread our manuscript again and corrected the typos.

Point 19: Using “healthy human” and “healthy control” are not appropriate as humans without BD may have other health conditions. It is instead better to refer to them as individuals with BD and individuals without BD.

Response: we have replaced these terms with "individuals with BD" and "individuals without BD" throughout our manuscript, including in figures and legends.

Point 20: The axis labels in supplement are too small to read.

Response: We have updated the front size in supplementary figures.

Point 21: Figure 1C labels cell types with numbers but it is unclear what these numbers map to. They should be updated to agree with the cell type labels in B.

Response: We have added the cell type labels in Fig. 1C.

Point 22: Line 96 should include the full name hematopoietic stem and progenitor cells before introduction the abbreviation HSPC.

Response: We have added the full name.

Line 96-99: “The scATAC-seq dataset, aligned using dimension reduction and graph-based clustering, yielded discrete cell clusters, primarily representing T (CD4/CD8) cells, monocytes, dendritic cells (DCs), T cells, natural killer (NK) cells, B lymphocytes, and Hematopoietic stem and progenitor cell (HSPC), with comparable cellular composition in scATAC-seq and scRNA-seq datasets (Fig. 1C).”

Point 23: In line 97 it is confusing to compare the ATAC-seq dataset to the RNA-seq dataset before the RNA-seq has been introduced, this should move to the RNA-seq section. This is also true for lines 256-257

Response: We have rewritten the sentences as suggested.

Line 96-99: “The scATAC-seq dataset, aligned using dimension reduction and graph-based clustering, yielded discrete cell clusters, primarily representing T (CD4/CD8) cells, monocytes, dendritic cells (DCs), T cells, natural killer (NK) cells, B lymphocytes, and Hematopoietic stem and progenitor cell (HSPC), with comparable cellular composition in scATAC-seq and scRNA-seq datasets (Fig. 1C).”

Line 270: “The results showed that NFkB family TF (RELB, NFkB1) showed high accessibility in BD patients.”

Point 24: In line 115 why is it unexpected that the DEGs aligned with the literature? I would think this is expected unless there is a perturbation or something else that I missed.

Response: We have corrected the sentence.

Line 124-127: “As expected, many differentially expressed genes (DEGs) in the six major immune cell types agreed with previous literature and our scATAC-seq dataset, such as IL7R for T cells, GNLY for NKs, S100A8 for monocytes, HLA-DQA1 for DC, MS4A1 for BC and GATA2 for HSPC^{2,3} (Fig. 1F).”

Point 25: Figure 1F seems to be missing pieces like the cell labels and the key. I also wonder if the z-score was performed correctly because I would expect the patterns to be much stronger for differentially expressed genes.

Response: We apologize for any confusion caused by the omission of cell labels and the key in Figure 1F. We have now included these essential elements in the updated version of the figure, which can be found below.

Figure 1F. Row-normalized single-cell gene expression heatmap of six main immune cell-type marker genes.

Regarding the range of the z-score in Figure 1F, we understand your expectation of stronger patterns for differentially expressed genes. However, we would like to clarify that our methodology uses a robust approach to identify significantly changed genes.

Point 26: Figure S2B shows CD4 Treg vs CD4 Naïve cells, the supplemental figure legend says CD8 Treg vs CD8 Naïve cells, and the text line 142 – 144 says CD4 Naïve vs CD8 T regs – which is correct for this comparison?

Response: We have corrected the typo and updated Fig. S3B. In the original figure legend for Figure S3B, there was a mistake in specifying the comparator groups. The correct comparator groups for the figure should have been " CD8 Treg vs. CD8 Naïve " cells, rather than " CD8 Treg vs. CD4 Naïve" cells, as stated in the text (lines 179-180).

Point 27: Lines 163-166 – The cited reference only discusses one of the genes mentioned. More references are necessary. Lines 225-226 also lack references for the provided genes.

Response: We have now added more references.

Line 173-179: “We observed that 14 genes were both upregulated in T cell subsets, including DUSP2⁴⁴, JUNB⁴⁵, IRF1⁴⁶, and DDIT4⁴⁵, suggesting T cells might be in proinflammatory state in BD patients. CD5⁴⁷, CD69⁴⁸, NFKBIA⁴⁹ were up-regulated in all the CD4 T cell subsets, suggesting CD4 T cell subsets were both highly activated in BD⁴⁸. In contrast, CD7⁵⁰, IL2RG⁵¹, IFITM1⁵², IFITM2⁵² were up-regulated in all the CD8 T cell subsets (Fig. S5C, S5D).”

Line 233-237: “As observed in the transcriptional data, all the B cell subsets in BD highly expressed interferon-stimulated genes IRF1⁴⁶, IFITM2⁵², and antigen processing and presentation-related molecules HLA-DQB1⁵³, and the cytokine IL2RG⁵⁴, as well as AP-1 family genes JUNB⁵⁵, MAPK signaling, and NF-κB signaling related genes DUSP1⁴⁴, NFKBIA⁴⁹ (Fig. 3G, Fig. S9C-F, Supplementary Data 4).”

Point 28: Line 194 is missing a reference to figure S5C.

Response: We have now added the reference to fig. S5C.

Line 206-211: “While we did not notice a significant change in the percentage of total NK cells (Fig. S8C, S8D), we noted significant transcriptional and epigenomic reconfiguration in all the NK subsets driven by up-regulated of several canonical NK cell activation genes (Fig. S8E-G, Supplementary data 4), including CD69⁴³ as well as interferon-stimulated genes (ISGs) IFITM2, IRF1, and ISG20 (Fig. 3C).”

Point 29: Figure 4k needs an x-axis label.

Response: We now have added the x-axis label on Fig. 4K.

Point 30: Figure S7D would be easier to interpret with a consistent color palette. For example, CM is Green, ActCM is Orange, NCM is Red rather than a consistent palette for high vs low.

Response: We have updated Fig. S11D (previous Fig. S7D) with a consistent color palette.

Point 31: Line 356 FigR is not cited. Please include <https://doi.org/10.1016/j.xgen.2022.100166>

Response: We have added the reference.

Line 396-398: “To address this, we have created well-integrated multi-omic data and used FigR⁵⁶ to deduce key transcriptional regulatory networks that are required for BD pathogenesis.”

Point 32: Line 567 indicates that a 5' 10x kit was used but the vin number is for a 3' kit.

Response: We have updated the vin number.

Line 631-633: “The Single-cell RNA libraries were prepared using the Chromium Single Cell 5 v2 Reagent (10x Genomics, 1000263) kit, following the manufacturer's instructions.

Reviewer #3 (Remarks to the Author):

Berchet disease is an inflammatory disease of unknown origin. This study aims to identify changes in chromatin accessibility and associated gene expression that correlate with disease pathogenesis. To achieve this, authors performed droplet-based scRNA-seq and scATAC-seq of peripheral blood mononuclear cells from patients and healthy individuals.

This study has several strengths worth noting. Firstly, a substantial number of patient samples were analyzed using state-of-the-art technology, which adds weight to the results. Additionally, although gene expression has already been reported, this is the first time scATAC analysis has been conducted, which represents a significant contribution to the field. Furthermore, the use of multiple analysis methods, including integrated analysis, to analyze single-cell data may offer valuable foundational knowledge for related fields.

Unfortunately, there are some weaknesses of this study. scRNA-seq analysis has already been reported previously. Moreover, because of the low correlation between scRNA-seq and scATAC-seq, this study did not significantly advance understanding of disease pathogenesis and progress mechanisms. Consequently, validation experiments using other approaches would be necessary to draw more robust conclusions.

Furthermore, while many of the figures shown contain cell cluster assignments, the most interesting comparative data between patients and healthy controls are secondary analysis data such as GO. Upset plots, for instance, make it challenging to interpret the data accurately due to the loss of information on expression levels. Therefore, figures that include quantitative information, such as Volcano plots, would be necessary.

Finally, the experimental results and their interpretations are not accurately described, and the text is challenging to understand in many areas, requiring improvement.

Response: Thank you for reviewing our manuscript. In response to your comments, we have conducted additional computational experiments to validate our findings and provide more robust support for our conclusions. Specifically, we have included additional data in Supplementary Data 2-4 to present all the DEG results between groups. This should provide a more comprehensive picture of the differences in gene expression between patient and control samples.

Regarding the low correlation between scRNA-seq and scATAC-seq, we agree that this presents a challenge for interpreting our results. To address this, we have focused on the interpretation of our results in the context of known biological processes and pathways, as reflected in the Gene Ontology (GO) terms assigned to each sample. We believe that this approach provides a more meaningful way of understanding the changes in gene expression that occur in response to disease, rather than simply comparing expression levels between groups.

In terms of figure presentation, we understand your concern about the difficulty of interpreting certain figures, particularly those that rely solely on qualitative assessments of gene expression. To address this, we have included additional figures that present quantitative information, such as volcano plots (Fig. S6A, Fig. S8G, Fig. S9D, Fig. S10F), to provide a more accurate and complete picture of our results.

We appreciate your feedback on the need for improvements in the experimental results

and their interpretations. We believe that these revisions will greatly enhance the readability and accessibility of our manuscript, and allow readers to focus more easily on the key findings and implications of our study.

Point 1: Fig. 1: Experimental design and cluster assignment. Some figure legends and figure contents are inconsistent. Please correct them.

Response: Thank you very much. We have corrected the typos in figure legends.

Figure 1A. Schematic highlighting design of single-cell multi-omics profiling of PBMCs from BD patients (scATAC: n=22; scRNA: n=23) and non-Behect's disease patients (n=8) in this study. Cells were then split and profiled using scATAC-seq and scRNA-seq for each condition.

Point 2: Fig. 1A: Number of samples for BD patients is 12?

Response: We have corrected the typo in Figure 1A.

Point 3: Are the colors in Fig. 1D and Fig. 1E the same? If so, please put the name of the cell type in

Response: We have added the name of the cell type in Figure 1F.

Point 4: line 118: Fig. 1E -> 1F?

Response: Thanks for pointing this out. We have renamed the Fig. 1E to Fig. 1F.

Point 5: Fig. 2D: TF footprint data does not practically show the binding of these TFs in both BD and HC patients. There are no clear footprints of these TFs. Consequently, I think there is no evidence for the role of AP-1 family in BD from this analysis.

Response: Based on our TF footprint analysis and statistical test, we believed that AP-1 family might have higher accessibility in BD patients. We have now added the statistical analysis on the TF footprint data. The statistical results are showed in supplementary

data 1.

Point 6: Line 165: Conclusion is unclear. Both CD4 and CD8 T cells are activated?

Response: Thank you for bringing this to our attention. We have updated the conclusion regarding the activation status of CD4 and CD8 T cells in BD patients. Our previous statement was unclear, and we have provided more specific information to address this point. Our analysis revealed that 14 genes (including DUSP2⁴⁴, JUNB⁴⁵, IRF1⁴⁶, and DDIT4⁴⁵) were upregulated in both CD4 and CD8 T cell subsets, indicating that both T cell populations may be in a proinflammatory state in BD patients. However, we also found that different T cell subsets had distinct profiles of gene expression. For example, CD5⁴⁷, CD69⁴⁸, NFKBIA⁴⁹ were upregulated in all CD4 T cell subsets, suggestive of high activation levels in these cells. In contrast, CD7⁵⁰, IL2RG⁵¹, IFITM1⁵², IFITM2⁵² were upregulated in all CD8 T cell subsets, indicating that these cells may also be activated but with a different profile compared to CD4 T cells. These findings are summarized in Fig. S5C and S5D.

Point 7: Line 176: which, 33, 34. Which following sentences are not mentioned?

Response: We have now added the missing part.

Line 188-191: “We observed that Th1 cells were involved in the NF-κB signaling, TNF signaling pathway. Th17 cells were involved in the response to the interferon-gamma pathway with interferon signaling related genes upregulated (ISG15, IFITM1, IFITM2) (Fig. S7B).”

Point 8: Fig. 3G: If Fig. 3G is a list of DEGs, it should be clearly stated in the text.

Response: We have now stated all the DEGs in Fig. 3G in the text and added all the list of DEGs in the supplementary data 2-4.

Line 238-239: “MB expressed higher RELA, BACH1, SPIB, JUN and JUND. DN2B showed higher expression on ISG20, TNFRSF13C and BTG1. (Fig. 3G, Supplementary Data 4)”

Point 9: Line 300 to 311: This section is a comparative analysis between monocytes. The relevance of this part to the last sentence is difficult to understand.

Response: We have rewritten the section to better connect the analysis of monocyte subsets with the overall message of the paper.

As mentioned earlier, we performed TF deviation analysis in monocyte subsets and discovered that each subset exhibited unique patterns of TF activity. Notably, ActCM and ActNCM displayed distinct variations in TF families related to NF-κB signaling and myeloid differentiation. ActCM showed enrichment in AP-1 family members, FOS, and JUN, which are typically associated with activation and maturation stages. In contrast, ActNCM had high activity of TFs involved in hematopoietic commitment and survival of monocytes, such as NR4A1 and NR4A2.

These findings highlight the diverse roles of monocyte subsets in BD and their potential contribution to the development of organ damage. By analyzing the transcriptomic landscape of these subsets, we can gain insights into the molecular mechanisms underlying their functional specialization and identify potential therapeutic targets for

treating BD.

Line 318-329: “We next performed TF deviation analysis on monocyte subsets, we noticed monocyte subsets exhibited deviated variations in different TF family members from homeostasis to the activated effector state (Fig. S11D). For example, both ActCM and ActNCM were associated with high levels of activity of TF involved in NF- κ B signaling signatures and myeloid differentiation, including REL, RELA, NFKB1, and NFKB2^{57,58} (Fig. S11D). Interestingly, ActCM had increased enrichment in TFs that represent activation and maturation stages, such as AP-1 family members, FOS, and JUN^{55,58,59}. In addition, CM was also enriched in the activation of AP-1 family TFs (Fig. 4L). Meanwhile, ActNCM showed high activity of TFs involved in haematopoietic commitment and survival of monocytes, including NR4A1 and NR4A2^{58,60} (Fig. 4L, Fig. S11D, S11E, Supplementary Data 1). Collectively, myeloid subsets maintained chromatin reprogramming and transcription changes that promote a rapid inflammatory response in BD.”

Point 10: Fig. 4I: There are two Fig. 4I.

Response: We have changed the second figure 4I to figure 4L.

Point 11: Fig. 5C: Fig. 5C needs a detailed explanation of the data and the process used to obtain it.

Response: Thank you for bringing up the need for a more detailed explanation of Fig. 5C and the methods used to obtain the data. We have now included additional information about the dataset and the analysis procedures in the Methods section and the figure legend.

In particular, we used a publicly available database of autoimmune and non-immune disorders from a previous study to calculate the enrichment of disease-related SNPs in 29 peripheral immune cell types using the g-chromVAR tool. We found that the majority of autoimmune diseases, such as BD, were significantly enriched in specific T cell subsets. On the other hand, type 1 diabetes was most strongly enriched in Th17 cells. Additionally, within the open chromatin regions of our broad cell subsets, g-chromVAR enrichment revealed significant T cell subsets, further supporting the previous study.

Moreover, we observed that T cells with cytotoxicity have been reported to contribute to BD pathogenesis in both skin and circulation, and our results show that these cells are indeed enriched in the disease-associated variants. Similarly, we saw trends of T cell enrichment for type 1 diabetes and asthma, although the significance did not reach statistical threshold. Importantly, most of the non-autoimmune disease GWAS were not apparent in the immune cell peaks, demonstrating the consistency of our results with previous studies.

While this aspect of our study was not the primary focus, we believe that our generated PBMC chromatin data could provide valuable insights into cellular-specific enrichment of human autoimmune disease heritability.

Line 361-375: “We used a publicly available database for autoimmune and non-immune disorders from previous study⁴ and calculated the enrichment of disease-related SNPs in 29 peripheral immune cell types using the g-chromVAR^{4,5}. The majority of the

autoimmune associations were strongly enriched for a corresponding trait association. For example, BD was significantly enriched in CD4 CTL, CD8TEM, DNT, and CD8TCM cells^{6,7}, while type 1 diabetes was most strongly enriched in Th17 cells (Fig. 5C)⁸. Within the open chromatin region of our broad cell subsets, g-chromVAR enrichment revealed significant T cell subsets, reinforcing the previous study^{6,7,9}. T cells with cytotoxicity have been reported to contribute to BD pathogenesis in both skin and circulation¹⁰. Additional trends of T cells enrichment are also observed here for type 1 diabetes and asthma, although not statistically significant. Most of these non-autoimmune disease GWAS were not apparent in the immune cell peaks, demonstrating GWAS enrichment was consistent with our expectations. Although not the focus of our current study, we observed that our generated PBMC chromatin data could provide cellular-specific enrichment of human autoimmune disease heritability.”

Line 616-626: “Pre-computed fine-mapped autoimmune-disease associated SNPs were downloaded from Farh et al⁴. We used the g-chromVAR⁵ algorithm to identify enrichment of disease variants in each cell type. In brief, the summary statistics we downloaded were converted to hg38 coordinates using the UCSC liftover tool (v377) and formatted for g-chromVAR5. The methodology of g-chromVAR was previously described in detail⁵. Briefly, g-chromVAR weights chromatin accessibility features by fine-mapped GWAS variants posterior probabilities and calculates the enrichment for each cell type feature intensity. We first binarized the scATAC-seq dataset matrix with one column per cell type. We then followed the recommended guidelines for GWAS enrichment using ‘computeWeightedDeviations’ with default parameters. We then applied the Mann-Whitney U test and the Benjamini-Hochberg procedure for multiple-testing correction to compute enrichment p values²⁰.”

Point 12: Fig. 5D: What was the reason for choosing this particular SNP? How about others? An explanation would be needed.

Response: Thank you for raising a concern about the choice of SNPs in Figure 5D. We understand the importance of providing clear justification for the selection of specific SNPs in the manuscript.

To address this point, we have added details about the rationale behind selecting the specific SNP in the KLRC4-KLRK1 enhancer region. We explained that this SNP was chosen due to its known association with BD risk and its location within an enhancer region that is specifically accessed in CD8 T cell and NK cell subsets. Moreover, we provided evidence that the peak-to-gene connection strength for this SNP is higher in non-BD individuals than in BD patients, suggesting a potential mechanism for how this variant contributes to the pathogenesis of BD.

Regarding the other SNPs in the GWAS hit list, we want to emphasize that our study focused on identifying the most strongly predicted disease-associated variants rather than exhaustively examining all possible SNPs. While there may be other potentially relevant SNPs in the GWAS list, our analysis prioritized the most well-established associates based on existing literature and our own findings.

However, we acknowledge that exploring the functionality of other GWAS hits could provide important insights into the molecular mechanisms underpinning BD.

Line 376-391: “We compiled a list of 66 index SNPs from Farh et.al⁴ representing GWAS hits for BD. We then identified all the SNPs in scATAC peaks and focused on their nearest genes. Furthermore, we calculated peak-to-gene connections for these gene locus using

ArchR (Fig. 5D, Fig. S13E-G). We noticed the rs201985743¹¹, that confers the risk of BD was in the KLRC4-KLRK1 enhancer region, which was opened in CD8 T cell subsets and NK cells subsets¹². This enhancer was highly accessible in NK and CD8 T subsets, not in T cells, B cells or monocytes, demonstrating NK and CD8 T specificity. In KLRC4-KLRK1 enhancer region showed statistically increasingly strong peak-to-gene linkages in non-BDs compared to BD patients (Fig. 5D, Fig. S13G). However, we did not notice other SNP loci showed stronger predictive linkages between BD and non-BDs. The low peak accessibility and gene expression in the BD state illustrates chromatin dynamic regulation in the KLRC4-KLRK1 locus. Since Killer cell lectin-like receptor subfamily (KLRC) regulates NK and CD8T function¹²⁻¹⁴, it is possible that this SNP contribute to the pathogenesis of BD by dysregulating KLRC4-KLRK1 in NK and CD8 T function. Therefore, our multi-omic integration analysis could provide predictive disease mechanisms that involve alterations in BD chromatin and gene regulatory regions.”

Point 13: Fig. 6: In the FigR analysis, a large number of TF-DORC associations are proposed. The original paper seems to have used TF-DORC associations with a regulation score threshold greater than 1 for the analysis. I assume that appropriate filtering will select more important TF-DORCs in this study as well.

Response: Thank you for pointing out the large number of TF-DORC associations in Fig. 6 and bringing to our attention the use of a regulation score threshold greater than 1 in the original paper. To address this issue, we have applied a similar filtering strategy to identify the most relevant TF-DORC associations in our study.

Specifically, we have used a regulation score threshold of $\text{abs}(\text{regulation score}) \geq 1$ to filter out weak associations and focussed on the top-ranked TF-DORC pairs with high absolute values of regulation scores. These filters helped us to narrow down the list of TF-DORC associations to only those that are most likely to play a crucial role in the regulation of HLA-DQB1 expression. Our updated analysis should provide a more refined view of the TF-DORC interactions that are most relevant to HLA-DQB1 regulation.

Reference:

1. Weirauch, M.T., *et al.* Determination and inference of eukaryotic transcription factor sequence specificity. *Cell* **158**, 1431-1443 (2014).
2. Shi, W., *et al.* Chromatin accessibility analysis reveals regulatory dynamics and therapeutic relevance of Vogt-Koyanagi-Harada disease. *Commun Biol* **5**, 506 (2022).
3. Hao, Y., *et al.* Integrated analysis of multimodal single-cell data. *Cell* **184**, 3573-3587 e3529 (2021).
4. Farh, K.K., *et al.* Genetic and epigenetic fine mapping of causal autoimmune disease variants. *Nature* **518**, 337-343 (2015).
5. Ulirsch, J.C., *et al.* Interrogation of human hematopoiesis at single-cell and single-variant resolution. *Nat Genet* **51**, 683-693 (2019).
6. Yang, J.Y., Park, M.J., Park, S. & Lee, E.S. Increased senescent CD8+ T cells in the peripheral blood mononuclear cells of Behcet's disease patients. *Arch Dermatol Res* **310**, 127-138 (2018).
7. Yasuoka, H., *et al.* Preferential activation of circulating CD8+ and gammadelta T cells in patients with active Behcet's disease and HLA-B51. *Clin Exp Rheumatol* **26**, S59-63 (2008).
8. Shao, S., *et al.* Th17 cells in type 1 diabetes. *Cell Immunol* **280**, 16-21 (2012).
9. Satpathy, A.T., *et al.* Massively parallel single-cell chromatin landscapes of human immune cell development and intratumoral T cell exhaustion. *Nat Biotechnol* **37**, 925-936 (2019).
10. Chang, L., *et al.* Single cell Transcriptome and T cell Repertoire Mapping of the Mechanistic Signatures and T cell Trajectories Contributing to Vascular and Dermal Manifestations of Behcet Disease. *bioRxiv*, 2022.2003.2022.485251 (2022).

11. Kirino, Y., *et al.* Genome-wide association analysis identifies new susceptibility loci for Behcet's disease and epistasis between HLA-B*51 and ERAP1. *Nat Genet* **45**, 202-207 (2013).
12. Bauer, S., *et al.* Activation of NK cells and T cells by NKG2D, a receptor for stress-inducible MICA. *Science* **285**, 727-729 (1999).
13. Guma, M., *et al.* The CD94/NKG2C killer lectin-like receptor constitutes an alternative activation pathway for a subset of CD8+ T cells. *Eur J Immunol* **35**, 2071-2080 (2005).
14. Kim, D.K., *et al.* Human NKG2F is expressed and can associate with DAP12. *Mol Immunol* **41**, 53-62 (2004).
15. Kucuksezer, U.C., *et al.* The Role of Natural Killer Cells in Autoimmune Diseases. *Front Immunol* **12**, 622306 (2021).
16. Karasneh, J., Gul, A., Ollier, W.E., Silman, A.J. & Worthington, J. Whole-genome screening for susceptibility genes in multicase families with Behcet's disease. *Arthritis Rheum* **52**, 1836-1842 (2005).
17. Kumasaka, N., Knights, A.J. & Gaffney, D.J. High-resolution genetic mapping of putative causal interactions between regions of open chromatin. *Nat Genet* **51**, 128-137 (2019).
18. Mumbach, M.R., *et al.* Enhancer connectome in primary human cells identifies target genes of disease-associated DNA elements. *Nat Genet* **49**, 1602-1612 (2017).
19. Broekema, R.V., Bakker, O.B. & Jonkers, I.H. A practical view of fine-mapping and gene prioritization in the post-genome-wide association era. *Open Biol* **10**, 190221 (2020).
20. Duong, T.E., *et al.* A single-cell regulatory map of postnatal lung alveologenesis in humans and mice. **2**, 100108 (2022).
21. Zheng, W., *et al.* Single-cell analyses highlight the proinflammatory contribution of C1q-high monocytes to Behcet's disease. *Proc Natl Acad Sci U S A* **119**, e2204289119 (2022).
22. Nanke, Y., Yago, T. & Kotake, S. The Role of Th17 Cells in the Pathogenesis of Behcet's Disease. *J Clin Med* **6**(2017).
23. Okubo, M., *et al.* Transcriptome analysis of immune cells from Behcet's syndrome patients: the importance of IL-17-producing cells and antigen-presenting cells in the pathogenesis of Behcet's syndrome. *Arthritis Res Ther* **24**, 186 (2022).
24. Tong, B., Liu, X., Xiao, J. & Su, G. Immunopathogenesis of Behcet's Disease. *Front Immunol* **10**, 665 (2019).
25. Crowell, H.L., *et al.* muscat detects subpopulation-specific state transitions from multi-sample multi-condition single-cell transcriptomics data. *Nat Commun* **11**, 6077 (2020).
26. Ritchie, M.E., *et al.* limma powers differential expression analyses for RNA-sequencing and microarray studies. *Nucleic Acids Res* **43**, e47 (2015).
27. Shen, L.J.R.P. GeneOverlap: An R package to test and visualize gene overlaps. *R Package* **3**(2014).
28. Fu, R., *et al.* clustifyr: an R package for automated single-cell RNA sequencing cluster classification. *F1000Res* **9**, 223 (2020).
29. Wang, R., Jaw, J.J., Stutzman, N.C., Zou, Z. & Sun, P.D. Natural killer cell-produced IFN-gamma and TNF-alpha induce target cell cytolysis through up-regulation of ICAM-1. *J Leukoc Biol* **91**, 299-309 (2012).
30. Katagiri, T., *et al.* JunB plays a crucial role in development of regulatory T cells by promoting IL-2 signaling. *Mucosal Immunol* **12**, 1104-1117 (2019).
31. Koizumi, S.I., *et al.* JunB regulates homeostasis and suppressive functions of effector regulatory T cells. *Nat Commun* **9**, 5344 (2018).
32. Klaus, S.J., *et al.* Costimulation through CD28 enhances T cell-dependent B cell activation via CD40-CD40L interaction. *J Immunol* **152**, 5643-5652 (1994).
33. Hemann, E.A., *et al.* Interferon-lambda modulates dendritic cells to facilitate T cell immunity during infection with influenza A virus. *Nat Immunol* **20**, 1035-1045 (2019).
34. Maney, N.J., Reynolds, G., Krippner-Heidenreich, A. & Hilkens, C.M.U. Dendritic cell maturation and survival are differentially regulated by TNFR1 and TNFR2. *J Immunol* **193**, 4914-4923 (2014).
35. Tewary, P., *et al.* Granulysin activates antigen-presenting cells through TLR4 and acts as an immune alarmin. *Blood* **116**, 3465-3474 (2010).
36. Stewart, A., *et al.* Single-Cell Transcriptomic Analyses Define Distinct Peripheral B Cell Subsets and Discrete Development Pathways. *Front Immunol* **12**, 602539 (2021).
37. Leylek, R., *et al.* Chromatin Landscape Underpinning Human Dendritic Cell Heterogeneity. *Cell Rep* **32**, 108180 (2020).
38. Yu, G., Wang, L.G. & He, Q.Y. ChIPseeker: an R/Bioconductor package for ChIP peak annotation, comparison and visualization. *Bioinformatics* **31**, 2382-2383 (2015).
39. Fang, Y., *et al.* The role of a key transcription factor PU.1 in autoimmune diseases. *Front Immunol* **13**, 1001201 (2022).
40. Barnabei, L., Laplantine, E., Mbongo, W., Rieux-Laucat, F. & Weil, R. NF-kappaB: At the Borders of

- Autoimmunity and Inflammation. *Front Immunol* **12**, 716469 (2021).
41. Xu, W.D., Pan, H.F., Ye, D.Q. & Xu, Y. Targeting IRF4 in autoimmune diseases. *Autoimmun Rev* **11**, 918-924 (2012).
 42. Park, S.H., *et al.* BATF regulates collagen-induced arthritis by regulating T helper cell differentiation. *Arthritis Res Ther* **20**, 161 (2018).
 43. Borrego, F., Robertson, M.J., Ritz, J., Pena, J. & Solana, R. CD69 is a stimulatory receptor for natural killer cell and its cytotoxic effect is blocked by CD94 inhibitory receptor. *Immunology* **97**, 159-165 (1999).
 44. Lang, R. & Raffi, F.A.M. Dual-Specificity Phosphatases in Immunity and Infection: An Update. *Int J Mol Sci* **20**(2019).
 45. Huang, Z., *et al.* Effects of sex and aging on the immune cell landscape as assessed by single-cell transcriptomic analysis. *Proc Natl Acad Sci U S A* **118**(2021).
 46. Forero, A., *et al.* Differential Activation of the Transcription Factor IRF1 Underlies the Distinct Immune Responses Elicited by Type I and Type III Interferons. *Immunity* **51**, 451-464 e456 (2019).
 47. Schuster, C., *et al.* CD5 Controls Gut Immunity by Shaping the Cytokine Profile of Intestinal T Cells. *Front Immunol* **13**, 906499 (2022).
 48. Maecker, H.T., McCoy, J.P. & Nussenblatt, R. Standardizing immunophenotyping for the Human Immunology Project. *Nat Rev Immunol* **12**, 191-200 (2012).
 49. Ma, P., *et al.* Immune Cell Landscape of Patients With Diabetic Macular Edema by Single-Cell RNA Analysis. *Front Pharmacol* **12**, 754933 (2021).
 50. Png, Y.T., *et al.* Blockade of CD7 expression in T cells for effective chimeric antigen receptor targeting of T-cell malignancies. *Blood Adv* **1**, 2348-2360 (2017).
 51. Chin, S.S., *et al.* T cell receptor and IL-2 signaling strength control memory CD8+ T cell functional fitness via chromatin remodeling. *Nature Communications* **13**, 2240 (2022).
 52. Gómez-Herranz, M., Taylor, J. & Sloan, R.D. IFITM proteins: Understanding their diverse roles in viral infection, cancer, and immunity. *Journal of Biological Chemistry* **299**, 102741 (2023).
 53. Zajacova, M., Kotrbova-Kozak, A. & Cerna, M. Expression of HLA-DQA1 and HLA-DQB1 genes in B lymphocytes, monocytes and whole blood. *Int J Immunogenet* **45**, 128-137 (2018).
 54. Le Floch-Ramondou, A., *et al.* Blockade of Common Gamma Chain Cytokine Signaling with REGN7257, an Interleukin 2 Receptor Gamma (IL2RG) Monoclonal Antibody, Protected Mice from Inflammatory and Autoimmune Diseases. *Blood* **140**, 473-474 (2022).
 55. Fontana, M.F., *et al.* JUNB is a key transcriptional modulator of macrophage activation. *J Immunol* **194**, 177-186 (2015).
 56. Kartha, V.K., *et al.* Functional inference of gene regulation using single-cell multi-omics. *Cell Genom* **2**(2022).
 57. Ichiyama, T., *et al.* NF-kappaB activation in peripheral blood monocytes/macrophages and T cells during acute Kawasaki disease. *Clin Immunol* **99**, 373-377 (2001).
 58. You, M., *et al.* Single-cell epigenomic landscape of peripheral immune cells reveals establishment of trained immunity in individuals convalescing from COVID-19. *Nat Cell Biol* **23**, 620-630 (2021).
 59. Behmoaras, J., *et al.* Jund is a determinant of macrophage activation and is associated with glomerulonephritis susceptibility. *Nat Genet* **40**, 553-559 (2008).
 60. Hanna, R.N., *et al.* The transcription factor NR4A1 (Nur77) controls bone marrow differentiation and the survival of Ly6C- monocytes. *Nat Immunol* **12**, 778-785 (2011).

REVIEWERS' COMMENTS:

Reviewer #1 (Remarks to the Author):

The authors have responded to all of my concerns and present a much more rigorous manuscript. Although the findings are largely descriptive, the integration of the datasets and large cohort size add novelty. Below are editorial changes that should be made to the manuscript to further improve figure resolution, readability, and in the flow of various figure panels.

1. Many of the figures are in fact now much higher resolution but need to be rasterized. In their current form they are vectorized and load slowly, especially in the supplemental figures. Also rasterizing the heatmap in Fig 1F would improve the resolution as now it is hard to visualize any positive signal with the blue dominant background.
2. Line 61 – “monocular” should be “mononuclear”.
3. Round numbers in lines 90-91 to nearest decimal place.
4. Add a Y-axis label for Fig 1C.
5. Line 381 – “not not” should be “but not”.
6. The panels in Fig 6 are out of order. Panel C and D should be swapped.
7. The panels in Fig S13 are out of order. For example G is placed above E and F. Also provide a genome plot example for the KLRC4-KLRK1 enhancer similar to the other two genes shown

Reviewer #2 (Remarks to the Author):

Thank you for your detailed response. I especially appreciate the comparison between Seurat and Muscat. I was a bit surprised by your findings as I often see many more genes from Seurat than from muscat. For your batch correction, I agree that it did not appear that correction was needed, it just wasn't clear to me in the first draft if a correction had been done or not. The harmony corrections do not need to be included but the uncorrected UMAPs and confusion matrix should be kept as well as the sentence indicating that no batch correction was required.

Thank you also for adding in more information on the heatmap in figure 1. With the scale, it is now clear why the patterns appear weak. There are a couple of cells that are much higher than the mean for a few genes – IGKV3-20, FCER1A. This makes the z-score scale go up to 10 making the rest of the pattern appear much weaker. This is not a problem and can only be seen if the scale is present. I believe the heatmaps that are made by default by ArchR and Seurat automatically floor the z-score values to -2.5 and 2.5 which is why many heatmaps look like the signal is stronger.

Overall, all of my concerns have been corrected. I do have a few comments

1. I believe there is a typo on line 74 “nominated potential TF activators drivers” I think should have only activators or drivers, not both.
2. Lines 376-391 – the authors indicate that there is a difference in the number of peak to gene linkages between BD and non-BD individuals. Is the peak at rs201985743 found in the peak to gene linkages in both sets of individuals? Is there a difference in the expression of KLRC4-KLRK1 in NK and T cell subsets between the BD and non-BD individuals? Including a violin plot showing gene expression (in the RNA) of the BD vs non-BD and indicating if there was a significant difference detected would be helpful. It would also be helpful to include tracks with the cell types of interest that have been separated by BD vs non-BD.
3. Line 552 – The command should be “cellranger-atac count”

Reviewer #3 (Remarks to the Author):

Thanks for responding to my comments. The authors addressed all my concerns.

Subject: Re-submission of the following revised manuscript
Ref.: Manuscript COMMSBIO-23-1159-B

Title: Single-cell chromatin accessibility and transcriptomic characterization of Behcet disease

Dear Editor and Reviewers,

Thank you for the positive comments and time. We have addressed the comments and have rephrased the manuscript and figures. Our detailed point-by-point responses can be found below. To distinguish our answers and reviewers' easily, under each comment below in *italics*, I have submitted our responses in **bold**.

Thank you so much for your time and consideration.

I am looking forward to hearing from you.

Best regards,
Yingfeng

Professor Yingfeng Zheng
Zhongshan Ophthalmic Center, Sun Yat-sen University
#7 Jinsui Road, Tianhe District, Guangzhou, P.R. China
Email: zhyfeng@mail.sysu.edu.cn
Phone: +86 13922286455

Responses to Reviewer Comments

Reviewer #1 (Remarks to the Author):

The authors have responded to all of my concerns and present a much more rigorous manuscript. Although the findings are largely descriptive, the integration of the datasets and large cohort size add novelty. Below are editorial changes that should be made to the manuscript to further improve figure resolution, readability, and in the flow of various figure panels.

Response: Thank you. We have improved the figure resolution, readability and in the flow of various figure panels.

1. Many of the figures are in fact now much higher resolution but need to be rasterized. In their current form they are vectorized and load slowly, especially in the supplemental figures. Also rasterizing the heatmap in Fig 1F would improve the resolution as now it is hard to visualize any positive signal with the blue dominant background.

Response: We have updated and rasterized the fig. 1f for better visualization. We also rasterized other figures to improve the loading speed.

Figure 1f. Row-normalized single-cell gene expression heatmap of six main immune cell-type marker genes. All data are aligned and annotated to hg38 reference genome.

2. Line 61 – “monocular” should be “mononuclear”.

Response: Thank you, we have fixed the typo.

Line 66: “Over recent decades, progress in single-cell sequencing technologies has enabled profiling of the genetic transcriptomics of peripheral blood mononuclear cells (PBMCs) and skin tissues from BD patients”

3. Round numbers in lines 90-91 to nearest decimal place.

Response: We have rounded the numbers in lines 90-91.

Line 93: “After stringent quality control filtration, a total of 152,704 cells of the scATAC-seq dataset and 272,113 cells of the scRNA-seq dataset were retained for downstream analysis, with an average of 8,810 unique nuclear fragments and an average of 14.5 in TSS enrichment for scATAC-seq-profiled cells, and an average of 2,042.9 UMIs for scRNA-seq-profiled cells (Fig. 1B, Fig. S1A-N).”

4. Add a Y-axis label for Fig 1C.

Response: Thank you, we have added a Y-axis label for Fig. 1C.

Figure 1c, Total number of six main immune cell types profiled passing quality control filtering for scATAC and scRNA-seq.

5. Line 381 – “not not” should be “but not”.

Response: Thank you, we have fixed the typo.

Line 393: “This enhancer was highly accessible in NK and CD8 T subsets, but not in T cells, B cells or monocytes, demonstrating NK and CD8 T specificity.”

6. The panels in Fig 6 are out of order. Panel C and D should be swapped.

Response: We have swapped the order of Fig. 6C and Fig. 6D.

7. The panels in Fig S13 are out of order. For example G is placed above E and F. Also provide a genome plot example for the *KLRC4-KLRK1* enhancer similar to the other two genes shown.

Response: We have updated Supplementary Figure 13, and provided genome plot for the *KLRC4-KLRK1* plot similar to the other two genes shown.

Supplementary Figure 13e. Cis-regulatory architecture in PBMCs (left panel: non-BD groups; right panel: BD groups): *KLRC4-KLRK1*. Only connections originating in the loci with peak-to-gene accessibility above 0.4 are shown.

Reviewer #2 (Remarks to the Author):

Thank you for your detailed response. I especially appreciate the comparison between Seurat and Muscat. I was a bit surprised by your findings as I often see many more genes from Seurat than from muscat. For your batch correction, I agree that it did not appear that correction was needed, it just wasn't clear to me in the first draft if a correction had been done or not. The harmony corrections do not need to be included but the uncorrected UMAPs and confusion matrix should be kept as well as the sentence indicating that no batch correction was required.

Thank you also for adding in more information on the heatmap in figure 1. With the scale, it is now clear why the patterns appear weak. There are a couple of cells that are much higher than the mean for a few genes – IGKV3-20, FCER1A. This makes the z-score scale go up to 10 making the rest of the pattern appear much weaker. This is not a problem and can only be seen if the scale is present. I believe the heatmaps that are made by default by ArchR and Seurat automatically floor the z-score values to -2.5 and 2.5 which is why many heatmaps look like the signal is stronger.

Overall, all of my concerns have been corrected. I do have a few comments

Response: Thank you for the positive comments and time. We did not conduct batch correction in the first draft. We have removed the figures of harmony correction and kept the uncorrected UMAPs and confusion matrix in

Supplementary Figure 1. We kept the sentence indicating that no batch correction was required. We also updated Fig. 1f for stronger signal.

Line 97: “We did not detect any potential batch effects in our datasets (Supplementary Figure 1d-g). Therefore, no batch correction method was applied in our further analysis.”

1. I believe there is a typo on line 74 “nominated potential TF activators drivers” I think should have only activators or drivers, not both.

Response: Thank you, we have fixed the typo.

Line 80: “Notably, we also nominated potential TF activators of chromatin accessibility and gene expression in BD.”

2. Lines 376-391 – the authors indicate that there is a difference in the number of peak to gene linkages between BD and non-BD individuals. Is the peak at rs201985743 found in the peak to gene linkages in both sets of individuals? Is there a difference in the expression of KLRC4-KLRK1 in NK and T cell subsets between the BD and non-BD individuals? Including a violin plot showing gene expression (in the RNA) of the BD vs non-BD and indicating if there was a significant difference detected would be helpful. It would also be helpful to include tracks with the cell types of interest that have been separated by BD vs non-BD.

Response: Thank you for bringing this to our attention. The peak at rs201985743 is found in the peak to gene linkages in both sets of individuals. We have shown the DEGs data of NK cell and T cell subsets in supplementary data. However, we did not notice a difference in the expression of KLRC1-KLRK1 in RNA level in NK and T cell subsets between BD and non-BD groups. We also computed the average gene expression of KLRC1-KLRK1 in the BD vs non-BD individuals in NK and T cell subsets and conducted the Wilcoxon rank sum test. The results were presented in Supplementary Data 9. We also did not notice significant difference in the expression of KLRC1-KLRK1, thus we did not intent to present these figures in the manuscript. The tracks with the cell types of interest separated by BD and non-BD were showed in Supplementary Figure 13e-g.

Supplementary Figure 13e. Cis-regulatory architecture in PBMCs (left panel: non-BD groups; right panel: BD groups): *KLRC4-KLRK1*. Only connections originating in the loci with peak-to-gene accessibility above 0.4 are shown.

Supplementary Figure 13f. Cis-regulatory architecture in PBMCs (left panel: non-BD groups; right panel: BD groups): *CCR3*. Only connections originating in the loci with peak-to-gene accessibility above 0.4 are shown.

Supplementary Figure 13g. Cis-regulatory architecture in PBMCs (left panel: non-BD groups; right panel: BD groups): *STAT4*. Only connections originating in the loci with peak-to-gene accessibility above 0.4 are shown.

Line 397: “There are no differences in the expression of *KLRC4-KLRK1* in NK and

T cell subsets in the scRNA-seq dataset (Supplementary Data 4, 7, 9)."

3. Line 552 – The command should be “cellranger-atac count”

Response: Line 566: “Subsequently, the scATAC-seq data reads were aligned to the GRCh38 (hg38) reference genome and quantified using the ‘cellranger-atac count’ function (10x Genomics, v.1.0.0).”

Reviewer #3 (Remarks to the Author):

Thanks for responding to my comments. The authors addressed all my concerns.

Response: Thank you for the positive comments and time.